# Rhesus Cytomegalovirus-encoded Fcγ-binding glycoproteins facilitate viral evasion from IgG-mediated humoral immunity

Claire E. Otero[1,2,8], Sophia Petkova[3,8], Martin Ebermann [3,8], Husam Taher[4], Nessy John[4], Katja Hoffmann [3], Angel Davalos[5], Matilda J. Moström [6], Roxanne M. Gilbride[4], Courtney R. Papen[4], Aaron Barber-Axthelm[4], Elizabeth A. Scheef[6], Richard Barfield[5], Lesli M. Sprehe[6], Savannah Kendall[6], Tabitha D. Manuel[6], Teresa Beechwood[4], Linh Khanh Nguyen[4], Nathan H. Vande Burgt[4], Cliburn Chan[5], Michael Denton[4], Zachary J. Streblow[4], Daniel N. Streblow[4], Alice F. Tarantal[7], Scott G. Hansen[4], Amitinder Kaur[6], Sallie Permar[1], Klaus Früh [4], Hartmut Hengel [3], Daniel Malouli [4,8] ✉ & Philipp Kolb [3,8] ✉

Human cytomegalovirus (HCMV) encodes four viral Fc-gamma receptors (vFcγRs) that counteract antibody-mediated activation in vitro, but their role in infection and pathogenesis is unknown. To examine their in vivo function in an animal model evolutionarily closely related to humans, we identified and characterized Rh05, Rh152/151 and Rh173 as the complete set of vFcγRs encoded by rhesus CMV (RhCMV). Each one of these proteins displays functional similarities to their prospective HCMV orthologs with respect to antagonizing host FcγR activation in vitro. When RhCMV-naïve male rhesus macaques were infected with vFcγR-deleted RhCMV, peak plasma DNAemia levels and anti-RhCMV antibody responses were comparable to wildtype infections of both male and female animals. However, the duration of plasma DNAemia was significantly shortened in immunocompetent, but not in CD4 + T cell-depleted animals. Since vFcγRs were not required for super-infection of rhesus macaques, we conclude that these proteins can prolong lytic replication during primary infection by evading virus-specific adaptive immune responses, particularly antibodies.

Cytomegaloviruses (CMVs) are a group of widespread, strictly species-specific β-herpesviruses that have been isolated from various rodent and primate species[1,2]. Primary infection of an immunocompetent, CMV-naïve host generally resolves asymptomatically followed by lifelong viral persistence and periodic reactivation[3]. Repeated exposure to viral antigens results in extraordinarily strong CMV-specific humoral and cellular immune responses that control, but do not clear the viral infection[3,4]. Conversely, uncontrolled human CMV (HCMV)

[1]Department of Pediatrics, Weill Cornell Medicine, New York, New York, USA. [2]Department of Pathology, Duke University, Durham, North Carolina, USA. [3]Institute of Virology, Medical Center, Faculty of Medicine, University of Freiburg, Freiburg, Germany. [4]Vaccine and Gene Therapy Institute, Oregon Health and Science University, Beaverton, Oregon, USA. [5]Department of Biostatistics and Bioinformatics, Duke University, Durham, North Carolina, USA. [6]Tulane National Primate Research Center, Tulane University, Covington, Louisiana, USA. [7]Departments of Pediatrics and Cell Biology and Human Anatomy, School of Medicine, and California National Primate Research Center, University of California, Davis, CA, USA. [8]These authors contributed equally: Claire E. Otero, Sophia Petkova, Martin Ebermann, Daniel Malouli, Philipp Kolb. ✉e-mail: maloulid@ohsu.edu; philipp.kolb@uniklinik-freiburg.de

infections in immunocompromised patients, such as transplant recipients can lead to severe morbidity and mortality[5]. Furthermore, congenital HCMV (cCMV) infections in neonates are the leading non-genetic cause of birth defects such as sensorineural hearing loss[6] and neurological deficits[7]. Thus, the development of a prophylactic HCMV vaccine for vulnerable populations has been designated a "Tier I priority" by the Institute of Medicine since 2000[8]. Yet to date, no HCMV vaccine approach has demonstrated sufficient efficacy in clinical trials to warrant licensure[9–11].

As productive HCMV infections are restricted to humans, we previously established a cCMV infection model in rhesus macaques (RMs) which allows for the examination of exploratory treatment and vaccine approaches in an evolutionarily closely related host[12]. In this model, we demonstrated that pre-exposure administration of highly concentrated IgG purified from plasma of RMs with potent anti-CMV IgG responses (hyper immunoglobulin, HIG) can limit the replication and congenital transmission of rhesus CMV (RhCMV)[13]. Although no consistent benefit of post-exposure HIG therapy has been demonstrated for cCMV infections across various human clinical trials, it appears that treatment dosage and pharmacokinetics are important parameters of efficacy and protection[14–18]. In line with these observations, we recently demonstrated that not neutralization, but IgG-Fc-gamma (Fcγ)-mediated anti-viral mechanisms by non-neutralizing IgG, specifically maternal plasma levels of antibody-dependent cellular phagocytosis (ADCP) and antibody-dependent cellular cytotoxicity (ADCC), correlate with a reduced risk of congenital infection[19,20]. These mechanisms are mediated by Fcγ receptors (FcγRs) expressed by almost all immune cells, including neutrophils, macrophages, and NK cells.

Several structurally unrelated viral IgG Fc-binding glycoproteins (vFcγRs) are found in the genomes of various herpesviruses, including the gE/gI glycoprotein complex encoded by herpes simplex virus (HSV)-1[21–23], gE encoded by Varicella zoster virus (VZV)[24–26] as well as multiple glycoproteins examined across different CMV species[27–32]. The HCMV genome encodes four vFcγRs, *RL11* (RL11 or gp34), *UL119/118* (UL119/118 or gp68), *RL12* (RL12 or gp95) and *RL13* (RL13 or gpRL13)[29,33]. While these proteins can interfere with host FcγR activation independently, we were furthermore able to demonstrate that UL119/118 and RL11 can also act synergistically[34,35].

We recently identified Rh05 (*RL11A*) as the first vFcγR encoded by RhCMV and demonstrated that Rh05 interferes with host FcγR activation in vitro. However, Rh05-deleted RhCMV did not display any defect in vivo, likely due to the expression of additional vFcγRs[28], which we here identify as Rh152/151 (*Rh152/151*) and Rh173 (*Rh173*). Deletion of all three vFcγRs from a previously described full-length (FL) RhCMV clone representative of a low-passage, primary isolate[2] resulted in the restoration of antibody-mediated host FcγR activation in vitro and significantly shortened plasma DNAemia in immunologically competent, but not CD4 + T cell-depleted, CMV-naïve RM. These results establish CMV-encoded vFcγRs as bona fide immune evasion proteins and suggest that targeting these proteins might improve the effectiveness of CMV vaccines and immunotherapies.

## Results

### Rh152/151 encodes a UL119/118 vFcγR ortholog that can block host FcγR function

Since HCMV encodes multiple vFcγRs, we hypothesized that RhCMV should similarly encode vFcγRs in addition to Rh05[28]. As the RhCMV open reading frame (ORF) *Rh152/151* represents a direct sequence ortholog of the known HCMV vFcγR UL119/118 (gp68), we examined the corresponding protein for its IgG binding activity[36]. The *Rh152/151* ORF in the fibroblast-adapted laboratory strain 68-1 contains a premature termination codon removing two YXXΦ sorting motifs[2,36] (Fig. 1A). Since our previous study found that C-terminal modifications of UL119/118 increased surface expression by decreasing

internalization and subsequent degradation[34], we compared IgG-binding of intact Rh152/151 obtained from FL-RhCMV[2] with strain 68-1-derived Rh152/151 and Rh152/151 in which the predicted transmembrane and cytoplasmic domains were replaced with that of human CD4 as described previously[34] (Fig. 1A). All three Rh152/151 constructs were able to bind IgG, with both the truncated and CD4-tail versions showing elevated IgG binding and reduced degradation similar to the previously described CD4-tail version of HCMV UL119/118[34]. Accordingly, when we co-expressed vFcγRs with rhesus CD4 using a T2A linker sequence to ensure equimolar expression levels[37] and tested their effect on FcγR activation by a rhesus CD4-specific rhesusized IgG, the truncated or CD4-tailed versions of Rh152/151 showed stronger antagonization of human CD16 (CD16A or FcγRIIIA F176V) activation compared to Rh152/151 from FL-RhCMV (Fig. 1B). Furthermore, similar to the effect of UL119/118 on human CD16[34], Rh152/151 was able to directly interfere with the binding of rhesus CD16 to immune complexes formed by anti-CD20 IgG (rituximab) on cells expressing both CD20 and Rh152/151 (Fig. 1C). In contrast, Rh05 was unable to inhibit CD16 binding (Fig. 1C), consistent with previous results obtained for HCMV RL11 (gp34)[34]. This suggests that RhCMV Rh152/151 represents a sequence and functional ortholog of the HCMV vFcγR UL119/118 (gp68).

### The highly polymorphic *RL11* family member Rh173 (*RL11T*) encodes for an IgG binding protein that shows conserved antagonization of host FcγR activation

In HCMV, three out of four identified vFcγRs are members of the *RL11* gene family. We therefore inserted every *RL11* family member annotated in the RhCMV genome into a p-IRES-eGFP vector and probed for IgG1 binding by transfected 293 T cells using flow cytometry (Fig. 2A). HA-staining of cells transiently transfected with a plasmid containing one of the RL11 family member not previously implicated in IgG binding demonstrated that Rh06 (*RL11B*), Rh08 (*RL11D*), Rh08.1 (*RL11E*) and Rh22 (*RL11L*) showed very low expression levels (Fig. 2A). However, as these genes either lack a signal peptide or a transmembrane domain in their original sequence, they are unlikely to encode for a vFcγR able to antagonize host FcγR activation on the cell surface. While Rh19 and Rh23 demonstrated below average expression levels, their protein expression was likely sufficient to detect Fcγ binding, yet only Rh173 displayed substantial IgG binding compared to the HCMV UL119/118 control. Intriguingly, Rh13.1 did not bind IgG although its HCMV ortholog RL13 was previously reported to do so[33]. *Rh173* is the only *RL11* gene located in the center of the genome in a locus homologous to the *ULb'* region in HCMV and not near the 5' terminus. On the amino acid level, this protein shows substantial sequence diversity across RhCMV isolates, a feature that it shares with the HCMV vFcγR RL12 (gp95)[38] (Supplementary Fig. S1). We therefore tested several Rh173 genotypes from different RhCMV isolates in an in vitro FcγR activation reporter assay[28,39] (Fig. 2B). Plasmids containing the *Rh173* ORFs were co-expressed with rhesus CD4 via a T2A linker sequence and transfected HeLa cells were incubated with a rhesus CD4-specific rhesusized mAb and human CD16 reporter cells. All examined Rh173 sequence variants showed a similar antagonization of CD16 activation, suggesting a conserved function independent of polymorphisms.

### RhCMV vFcγRs are expressed with similar kinetics, localize to subcellular vesicular structures and are integrated into newly formed virions

To characterize RhCMV vFcγRs in the context of infection, we constructed an FL-RhCMV recombinant devoid of all three vFcγR. We additionally replaced the *Rh13.1* ORF with a SIVgag antigen which has the dual effect of stabilizing the virus genome during in vitro propagation while allowing us to specifically monitor immune responses to the heterologous antigen[2,40] (Fig. 3A). Using FL-RhCMVΔRh13.1/SIVgag

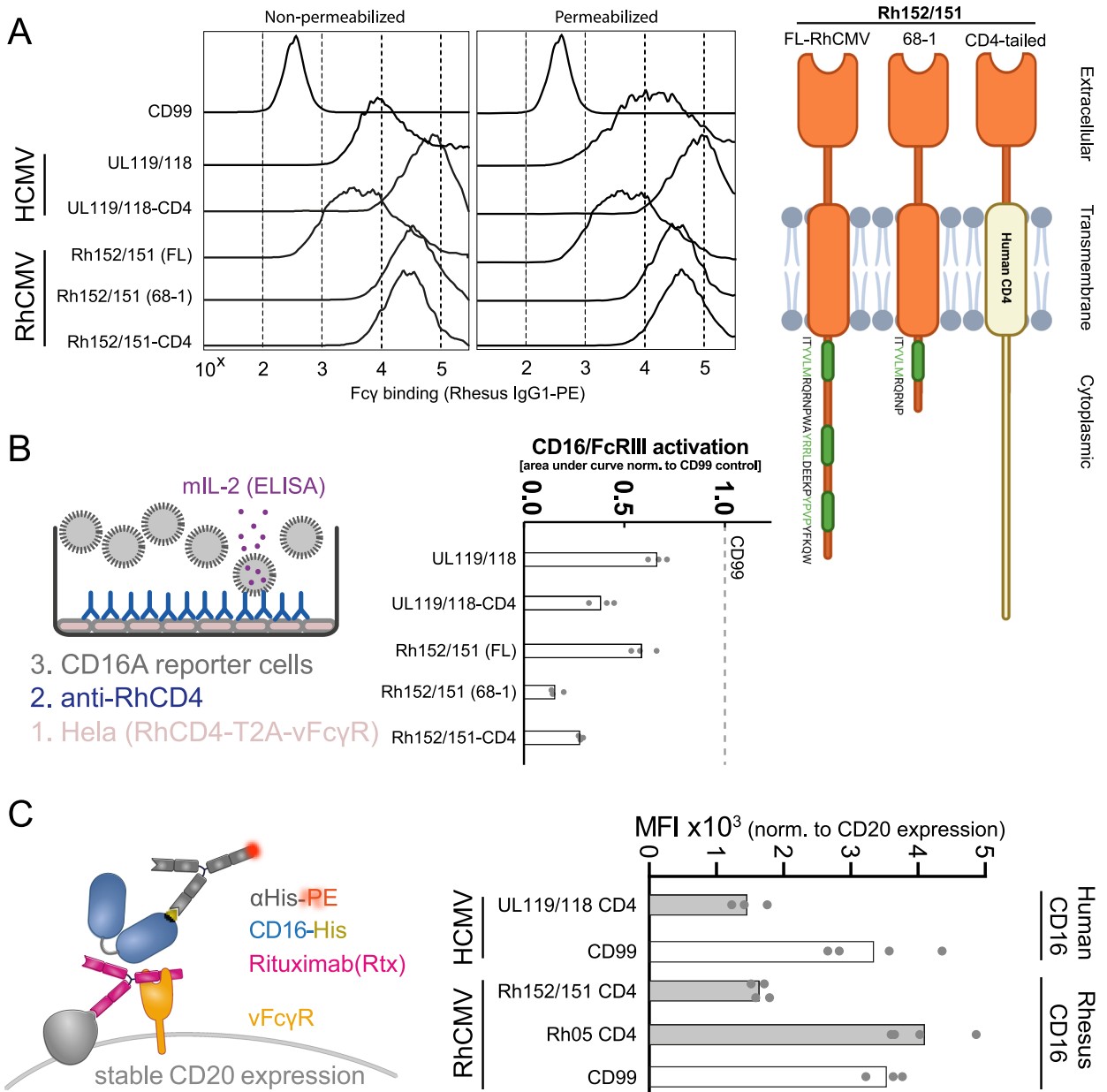

**Fig. 1 | Rh152/Rh151 binds IgG and antagonizes host FcγR activation by blocking receptor engagement. A** 293 T cells were transfected with a pIRES-eGFP plasmid expressing the indicated UL119/118 or Rh152/Rh151 constructs or human CD99. Cells were probed for binding of PE-conjugated rhesusized IgG1 by flow cytometry with or without permeabilization. Polycistronic GFP expression was used to gate on transfected cells. The cartoon illustrates the orientation and configuration of the Rh152/Rh151 constructs in the cellular membrane. Extracellular and cytoplasmic termini of the proteins are indicated, and the modifications to the YXXΦ sorting motifs are highlighted in green. Created in BioRender. Kolb, P. (2024) https://BioRender.com/w36j646. **B** HeLa cells recombinantly expressing the RhCD4 target antigen and the indicated vFcγRs in equimolar amounts from a T2A-linked fusion protein were incubated with graded amounts of RhCD4-specific

rhesusized IgG1 and tested for human CD16 activation using a cell-based reporter assay[34]. Symbols show area under curve (AUC) values from independent experiments normalized to activation in the presence of a non-Fcγ binding glycoprotein control (human CD99). Bars show the mean of independent experiments. **C** 293 T cells stably expressing human CD20 were transfected with the indicated vFcγRs or human CD99, incubated with the anti-CD20 antibody Rituximab, and probed for rhesus or human CD16 binding via flow cytometry. The experimental setup is shown in detail on the left. Symbols show the mean fluorescence intensity (MFI) of independent experiments normalized to CD20 expression following transfection. Bars show the mean of independent experiments. Source data are provided as a Source Data file.

as the parental recombinant we sequentially deleted *Rh05, Rh152/151* and *Rh173* with the corresponding mRNAs being no longer detectable in infected fibroblasts (Fig. 3B). Deletion of the vFcγR did not affect virus replication in primary rhesus fibroblasts (RFs) compared to FL-RhCMV (Fig. 3C). Similarly, it also did not alter the ability of FL-RhCMV to infect epithelial cells in vitro, contrary to the deletion of Rh157.5 (UL128) and Rh157.4 (UL130), two subunits of the pentameric complex (PC) required for entry into most non-fibroblast cells[41] (Fig. 3D).

We subsequently examined vFcγR mRNA expression levels in RFs infected with FL-RhCMV at an MOI of 3 over a 48 h interval. Expression levels of the *Rh156* (UL123, IE1), *Rh189* (US11), and *Rh137* (UL99) viral mRNAs expressed with representative immediate early (IE), early (E), and late (L) kinetics, respectively were included as controls[42]. We found that RhCMV vFcγRs were expressed with early-late (E-L) kinetics (Fig. 3E), matching their prospective HCMV encoded orthologs[29].

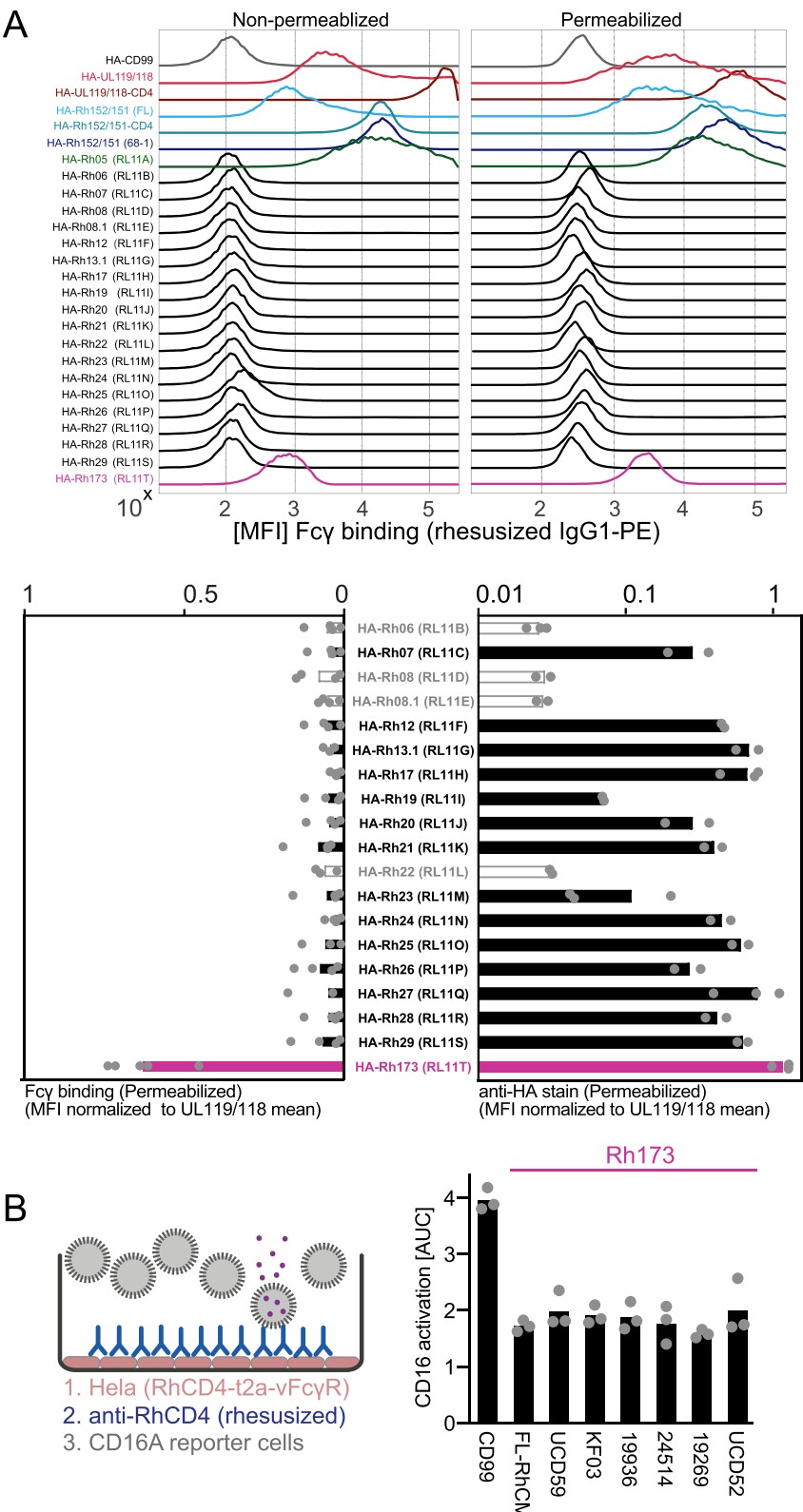

To track subcellular localization of vFcγRs in the context of viral infection, we generated a recombinant FL-RhCMV carrying a SIVgag transgene in the Rh13.1 locus and encoding C-terminally HA-tagged Rh05 and V5-tagged Rh173 (Fig. 3A). Rh152/151 was not epitope-tagged as a mAb[43] originally identified in a screen of RhCMV-specific mouse hybridoma cell lines recognized Rh152/151 (Supplementary Fig. S2) as well as Cynomolgus CMV (CyCMV) Cy152/151 (Supplementary Fig.

S3A), although it is unable to neutralize Rh152/151-mediated antagonization of host CD16 activation (Supplementary Fig. S3B). We infected telomerized RFs (tRFs) with the recombinant FL-RhCMVΔRh13.1/SIVgag/Rh05-HA/Rh173-V5 at an MOI of 1 and monitored the subcellular localization of each vFcγR by immunofluorescence analysis (IFA). We excluded non-specific binding of secondary antibodies by vFcγRs (Supplementary Fig. S4). At 24 h post-

**Fig. 2 | Rh173 binds IgG and antagonizes FcγR activation. A** 293 T cells transfected with pIRES-eGFP plasmids providing a Tapasin signal peptide and an N-terminal HA-tag encoding either an established vFcγR, an RhCMV *RL11* family member or the human CD99 protein as a negative control were probed for binding of PE-conjugated rhesusized IgG1 with or without permeabilization via flow cytometry. The presented histograms show a full panel of vFcγRs and a CD99 control to illustrate IgG binding ranges. The bar graphs focus on RL11 family members without known IgG binding. Symbols show the mean fluorescence intensity (MFI) of independent experiments for IgG-binding (left panel) or MFI of protein expression using the N-terminal HA-tag (right panel), both normalized to UL119/118 signals. Bars show the mean of independent experiments. Empty gray bars represent RL11 family members that lack a predicted signal peptide and/or transmembrane domain in their original sequence. **B** HeLa cells were transfected with plasmid pIRES-eGFP expressing a T2A-linked fusion protein of the RhCD4 target antigen, and the indicated Rh173 sequence variants from published RhCMV sequences. The transfectants were incubated with graded amounts of RhCD4-specific rhesusized IgG1 and tested for human CD16 activation using a cell-based FcγR activation reporter assay. Equal transfection efficiency was monitored via polycistronic GFP expression. A non-Fcγ binding glycoprotein control (CD99) served as control. Symbols show mean area under curve (AUC) values of independent experiments performed in technical replicates. Bars show the mean of independent experiments. Source data are provided as a Source Data file.

infection (hpi), all three vFcγR displayed a punctate vesicular staining pattern consistent with type I transmembrane glycoproteins located in subcellular vesicular structures (Fig. 3F and Supplementary Fig. S5). Co-staining of the vFcγRs indicated distinct localization of Rh05 whereas the localization of Rh152/151 and Rh173 seemed to overlap at least partially. Co-staining with markers for specific cellular compartments indicated that, in addition to EEA1 positive early endosomes, Rh152/151 and Rh173 showed co-localization with the Golgi-marker Rab6, whereas Rh05 partially co-localized with lysosomal and autophagosomal markers LAMP-1 and LC-3 (Fig. 3G and Supplementary Fig. S6). These staining patterns are consistent with vFcγRs being endocytosed and ultimately degraded in lysosomes at early times of infection. At 48 hpi, both Rh152/151 and Rh173 localized to perinuclear structures in the cell that co-stained with Rab6 and likely represent the viral assembly compartment (VAC) (Supplementary Figs. S5, S7).

The presence of vFcγRs in the VAC suggests that they are incorporated into virions during viral assembly which is consistent with mass spectrometry results published for strain 68-1[36]. We thus gradient-purified virions from the supernatant of RFs infected with FL-RhCMVΔRh13.1/SIVgag/Rh05-HA/Rh173-V5, FL-RhCMVΔRh13.1/SIVgag or triple deleted FL-RhCMVΔRh13.1/SIVgag/ΔΔΔ. All vFcγR was detected in immunoblots of purified virions from FL-RhCMVΔRh13.1/SIVgag/Rh05-HA/Rh173-V5 whereas they were absent from FL-RhCMVΔRh13.1/SIVgag/ΔΔΔ (Fig. 3H). Because Rh152/151 was detected directly with a mAb as described above, it was also found in virions from FL-RhCMVΔRh13.1/SIVgag. Since all vFcγR represent type I transmembrane proteins, they are likely inserted into the virion envelope during assembly, matching published results for HCMV[44].

## RhCMV encoded vFcγRs antagonize host FcγR activation in vitro
Our data suggest that RhCMV Rh152/151 is a clear ortholog of HCMV UL119/118 (gp68) and that Rh05 and Rh173 might share a conserved mechanism and role with RL11 (gp34) and RL12 (gp95) respectively (Fig. 4A). To more closely examine this functional similarity, we performed a side-by-side comparison of these proteins with respect to their ability to antagonize human CD16 activation upon transient expression using an FcγR activation assay (Fig. 4B). The observed levels of antagonization support the conclusion that the identified proteins represent orthologs with functional conservation across all RhCMV and HCMV encoded vFcγRs. To subsequently examine their ability to counteract host FcγR activation in the context of infection, we deleted individual vFcγRs or all three of them combined from a recombinant RhCMV strain 68-1 (68-1 R) in which ORF *Rh152/151* had been repaired by reversing the single nucleotide change resulting in a premature termination codon (Fig. 4C). Immunoprecipitations of IgG binding proteins using fully rhesusized IgG (anti-RhCD4, as above) from metabolically labeled tRFs infected with each of these recombinants confirmed expression of the expected vFcγRs (Fig. 4D). Deglycosylation using Endo H or PNGase F further confirmed RhCMV Rh05, Rh152/151 and Rh173 to be transmembrane glycoproteins that are processed via the ER-Golgi similar to HCMV vFcγRs[29] and indicates the absence of any further unidentified RhCMV encoded IgG binding

glycoproteins. Next, infected RFs were incubated with plasma pooled from eight different RhCMV-seropositive RM followed by incubation with FcγR-reporter cells expressing single human or rhesus FcγRs associated with ADCP or ADCC (Fig. 4E). Compared to RhCMV containing all three vFcγRs, an increased stimulation of all human and rhesus FcγRs was observed upon infection with a RhCMV recombinant lacking all three vFcγRs, consistent with them being able to antagonize all host FcγRs mediating ADCP or ADCC. This result also indicates that the FcγR activity is conserved between humans and RM consistent with previous IgG binding studies[45]. The deletion of either Rh05 or Rh152/151 alone resulted in a complete loss of antagonization, whereas the removal of Rh173 alone only marginally increased stimulation, which implies cooperation between Rh05 or Rh152/151 similar to HCMV UL119/118 and RL11[34]. Of note, the inhibitory effect on human and rhesus CD64, the only high-affinity FcγR and main mediator of ADCP, was especially dependent on the presence of all three vFcγRs. These observations are consistent with previous observations regarding HCMV[35] and are in line with the synergistic mechanisms postulated between individual HCMV vFcγRs[34].

## A RhCMV mutant deleted for all three vFcγRs is cleared prematurely from the circulation during primary infection in CMV naïve RM
To determine the role of vFcγRs on viral replication in vivo, we infected four RhCMV-naïve, male RM intravenously with $1 \times 10^6$ PFU of the FL-RhCMVΔRh13.1/SIVgag recombinant deleted of all three vFcγRs (ΔΔΔ) and monitored the development of plasma DNAemia over time by qPCR. As controls, we infected three RhCMV-seronegative male RM with the parental FL-RhCMVΔRh13.1/SIVgag. We included historical results of 12 RhCMV-seronegative, pregnant female RM infected intravenously with a single injection of $1 \times 10^6$ PFU of RhCMVΔRh13.1/SIVgag and $1 \times 10^6$ PFU RhCMV UCD52 each around the end of the first and the beginning of the second trimester. Viral genome copy numbers were detectable in all animals shortly after inoculation, reaching peak levels around 7 days post-infection (dpi) (Fig. 5A). However, the duration of the plasma DNAemia in animals infected with the vFcγR-deletion mutant was significantly shortened compared to FL-RhCMV (median time to control of viral infection 59.5 days versus 31.5 days post-infection) (Log-rank $p = 0.031$, Fig. 5B), and this finding maintained statistical significance in sensitivity analyses comparing the animals infected with FL-RhCMVΔΔΔ to only the male control animals which received FL-RhCMV (Log-rank $p = 0.029$, Supplementary Fig. S8 and Supplementary Table S2). Viral clearance from plasma coincided with peak humoral responses of antibodies binding to whole virions and specific viral glycoproteins (gB, PC) (Fig. 5C), as well as antibodies mediating viral neutralization, ADCP or ADCC (Fig. 5D). Kinetics of the RhCMV-specific antibody responses were indistinguishable from RM inoculated with the parental recombinant. Together, these results suggest that vFcγR-deletion does not affect the induction and specificity of the humoral immune responses, yet renders the virus more susceptible to antibody-mediated clearance. Intriguingly, we no longer observed any premature clearance of the (ΔΔΔ) deletion virus from the plasma of RhCMV-naïve, pregnant RM after CD4 + T-cells depletion

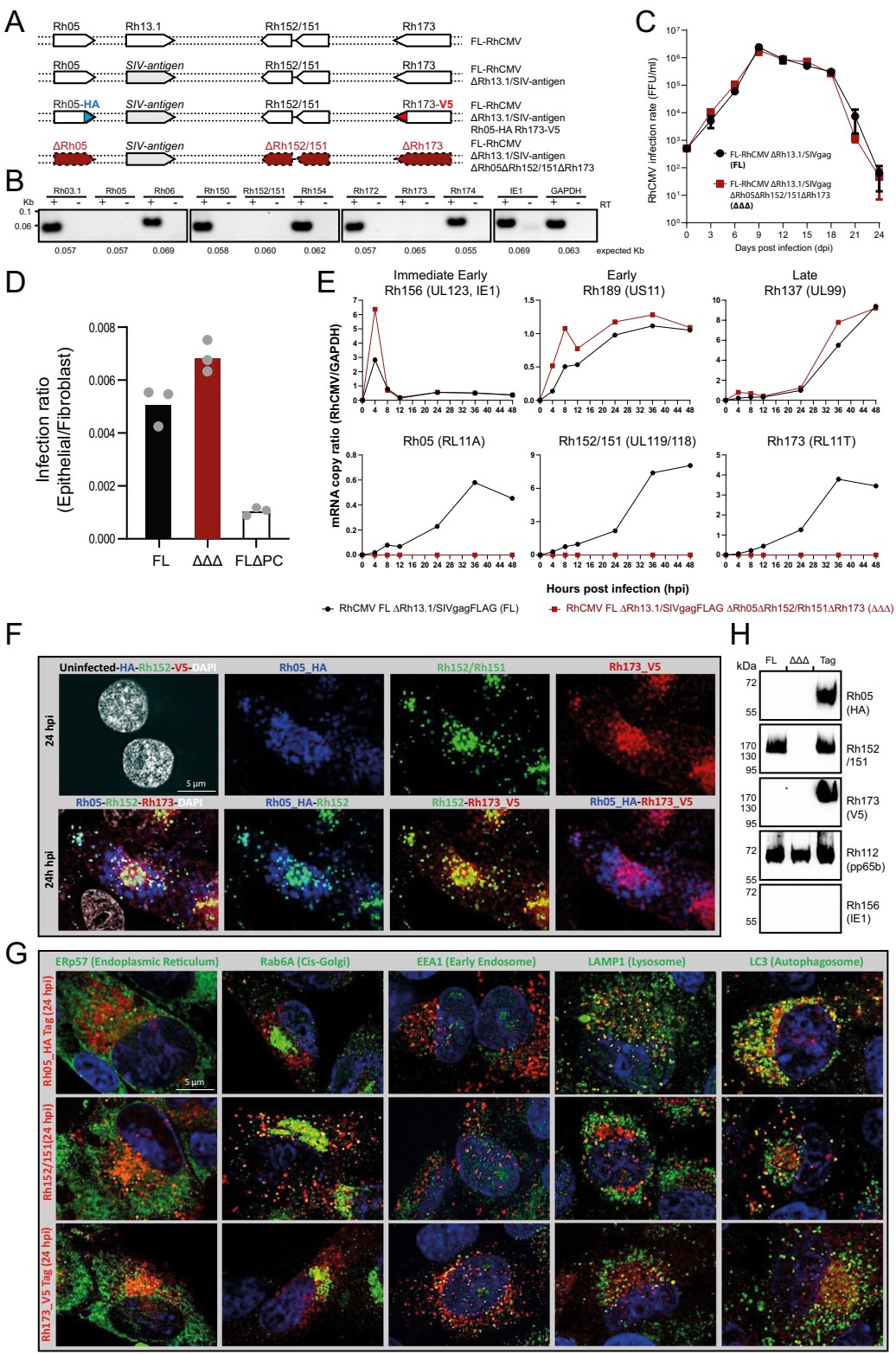

verified by flow cytometry one week prior to inoculation, which was confirmed by flow cytometry (Fig. 5E and Supplementary Fig. S9). Since depletion of CD4 + T cells leads to a delay in RhCMV-specific humoral immunity following inoculation[12], this result further supports a likely role of vFcγRs in counteracting the antibody response. Lastly, we investigated whether deletion of the vFcγRs would preclude superinfection of RhCMV seropositive RM. When we inoculated two immunocompetent, RhCMV seropositive animals with FL-RhCMVΔΔΔ carrying a SIV-5'Pol antigen in the *Rh13.1* locus we observed strong and

lasting SIV-5'Pol-specific CD4 + and CD8 + T-cell responses that developed shortly after inoculation, indicating that superinfection in the absence of humoral immune evasion genes is possible (Fig. 5F). In contrast, we previously reported that RhCMV recombinants devoid of viral gene products involved in evading MHC class-I mediated antigen presentation to CD8 + T cells lose their ability to superinfect RhCMV seropositive RM[40]. Taken together, these results show that vFcγRs support primary infection of CMV-naïve hosts by counteracting antibody-mediated clearance from the circulation.

**Fig. 3 | RhCMV vFcγRs are early-late gene products that are non-essential for growth in vitro and locate to sub-cellular vesicular structures in infected cells.** **A** Graphical overview of FL-RhCMV-derived recombinants, showing vFcγRs either deleted (highlighted in red) or fused to HA or V5 epitope tags (blue/red). Deleted vFcγRs are annotated in red. **B** vFcγR-encoding or neighboring gene mRNA expression was assessed by RT-PCR. Rhesus fibroblasts (RFs) infected with the triple vFcγRs deletion mutant (ΔΔΔ) at MOI = 5 were harvested at 36 hpi. cDNA was produced by reverse transcription, and PCRs were performed using primers targeting vFcγR-encoding or neighboring ORFs. Control samples without RT, RhCMV IE1, and GAPDH were included as controls. **C** Multistep growth curves on RFs compared replication kinetics of vFcγR-deleted (ΔΔΔ) and parental FL-RhCMV. RFs infected at MOI = 0.01, with viral titers measured as FFU/ml at indicated times. Data represent means ± SD (*n* = 3). **D** RFs and rhesus retinal pigment epithelial cells (RPEs) were infected with serial dilutions of FL-RhCMV, ΔΔΔ, or a pentameric complex (PC)-deleted mutant. Infection rates were assessed by IFA for pp65b, expressed as relative ratios of infected epithelial cells to fibroblasts. Symbols show mean values of technical replicates. Bars show means of independent experiments.

**E** vFcγRs expression kinetics were monitored in RFs infected with FL-RhCMV or ΔΔΔ (MOI = 5). Total mRNA harvested at multiple time points was analyzed by qRT-PCR. Viral transcripts were normalized to GAPDH. *Rh156* (UL123, IE1), *Rh189* (US11), and *Rh137* (UL99) served as kinetic controls for IE, E, and L classes. **F** Co-localization of vFcγRs was analyzed by infecting RFs with FL-RhCMV/Rh05-HA/Rh173-V5 (MOI = 1). Cells were stained for HA, Rh152/151, and V5. Representative single-cell views. Similar results were obtained in four independent experiments. Uncropped images and analysis at 48 hpi are shown in Supplementary Fig. S5. Secondary antibody control is shown in Supplementary Fig. 4. **G** Subcellular localization of vFcγRs was assessed using organelle markers (ER, Golgi, early endosome, lysosome, autophagosome). RFs infected with FL-RhCMV/Rh05-HA/Rh173-V5 showed specific co-localization patterns. Uncropped images, uninfected cells are shown in Fig. S6. An analysis at 48 hpi is shown in Fig. S7. Similar results were obtained in four independent experiments. **H** Virions of FL-RhCMV, ΔΔΔ, or FL-RhCMV/Rh05-HA/Rh173-V5 were purified. Immunoblots confirmed Rh05, Rh152/151, and Rh173 expression, with pp65 and IE serving as controls. Source data are provided as a Source Data file.

## Discussion

We have identified and characterized the full set of vFcγRs encoded by RhCMV including assessing their interference with antibody function in vivo in a highly relevant animal model for HCMV. In addition to Rh05, which we previously identified, we now show that Rh152/151 and Rh173 also bind to host IgG and interfere with host FcγR activation by immune complexes. *Rh152/151* was identified through sequence homology with *UL119/118*. Remarkably, clear sequence orthologs of this gene can be found in all examined old and new world non-human primate (NHP) CMVs and a non-spliced UL119/118-ortholog also displaying IgG-binding in vitro has even been identified in guinea pig CMV (GPCMV)[27], indicating that the ancestral gene predating all of these modern relatives was likely acquired by a common ancestor. Like HCMV UL119/118 (gp68), we observed that Rh152/151 is able to directly interfere with the binding of host FcγRs to antibody complexes, indicating that they bind a similar region on IgG (CH2-CH3 interdomain).

A systematic in vitro screen of all *RL11* family members encoded by RhCMV for IgG binding identified Rh173 as an additional vFcγR, while RhCMV Rh13.1, the ortholog of HCMV RL13 was unable to bind Fcγ, even though both proteins restrict virus replication in vitro and are hence selected against[33]. *Rh173* exhibits a high degree of sequence variability across RhCMV isolates consistent with continuous immune pressure, but despite this diversity, all tested sequence variants demonstrated equivalent vFcγR activity. Moreover, the absence of any additional IgG binding glycoproteins in an IgG precipitation experiment using RFs infected with RhCMV 68-1 devoid of Rh05, Rh152/151, and Rh173 implies that we have likely identified all vFcγR glycoproteins encoded by this virus.

The here characterized vFcγRs are expressed with similar E/L kinetics during virus replication, but their subcellular localization in infected cells varies significantly. Yet, deletion of individual vFcγRs revealed that both Rh05 and Rh152/151 seem to interfere with host FcγR activation cooperatively, similar to previously reported results for their HCMV orthologs[34]. This observation indicates that these proteins might interact transiently.

Deletion of the full set of vFcγRs did not affect replication or tropism in vitro, yet it eliminated viral interference with host FcγR activation. These results indicate that RhCMV vFcγRs do not seem to play a significant role in viral entry, replication, or release in tissue culture. This conclusion is furthermore supported by the observation that no difference in peak plasma DNAemia was detected between vFcγR-deleted FL-RhCMV and vFcγR-intact FL-RhCMV following infection of seronegative RM. Therefore, RhCMV vFcγRs are likely not required to overcome the immediate, innate host response to infection in vivo. This is in contrast to viral proteins such as Rh159 (UL148) which prevents the activation of NK cells through sequestration of NK cell activating ligands for NKG2D in virally infected cells[46].

Interestingly, the murine CMV vFcγR m138 also downregulates NKG2D-ligands and, as a result, affects viral replication in vivo, even before the onset of adaptive immune responses[47–51]. In contrast, no additional functions have so far been identified for primate CMV vFcγRs.

Perhaps unexpectedly, deletion of all vFcγRs from FL-RhCMV did not preclude superinfection of RhCMV seropositive RM as indicated by the development of de novo T cell responses to the inserted SIV antigen. These data are consistent with prior results showing that Rh05 is not required for reinfection[28]. In contrast, viral evasion of CD8 + T cells is essential to overcome pre-existing immunity[52]. Thus, it appears that evasion of humoral immunity may be a lesser requirement to overcome compared to T cell immune evasion of preexisting immune responses during superinfection. Previous studies analyzing the MHC-I ligandome in HCMV infected cells did not identify vFcγR-derived peptides as immuno-dominant epitopes[53], in line with our observation that the removal of vFcγRs did not increase the magnitude but lead to a shorter duration of DNAemia. However, as vFcγRs antagonize host FcγR activation, this could potentially affect host FcγR mediated cross-presentation of peptides[54], an aspect of vFcγR immune evasion that remains to be addressed experimentally.

The deletion of vFcγRs did not affect the initial phases of primary infection. However, there was a significant impact on the clearance of viral genomes from the circulation that coincided with the development of a robust anti-viral antibody response. Notably, this analysis does have limitations in the small number of only male animals infected with vFcγR-deleted FL-RhCMV and the inclusion of both male and female primates infected with different formulations of vFcγR-intact RhCMV. However, the effect remains statistically significant when comparing only male animals. This difference between vFcγR-intact and -deleted FL-RhCMV was not observed upon deletion of CD4 + T cells from CMV naïve RM prior to inoculation. Since CD4 + T cells are required for the timely development of antibody responses and depletion of CD4 + T cells resulted in a delay of RhCMV-specific antibody responses[12], this result strongly suggests that a vFcγR-deleted RhCMV is more susceptible to antibody control than wildtype RhCMV, consistent with a role in evading the humoral response. In future experiments, we plan to compare infection of B-cell depleted animals which is expected to further support this immune evasion function.

Targeting vFcγRs therapeutically, e.g., through antibodies that block their Fcγ-binding activities, could potentially increase the ability of anti-viral antibodies to control viral replication. Previously, we demonstrated that pre-treatment of pregnant females with HIG from RhCMV-positive animals prevented congenital transmission of RhCMV in immunosuppressed, RhCMV-seronegative animals[12,13]. However, this protective effect was only observed when the antibodies were isolated from donors with potent plasma virus-neutralizing activity. The results

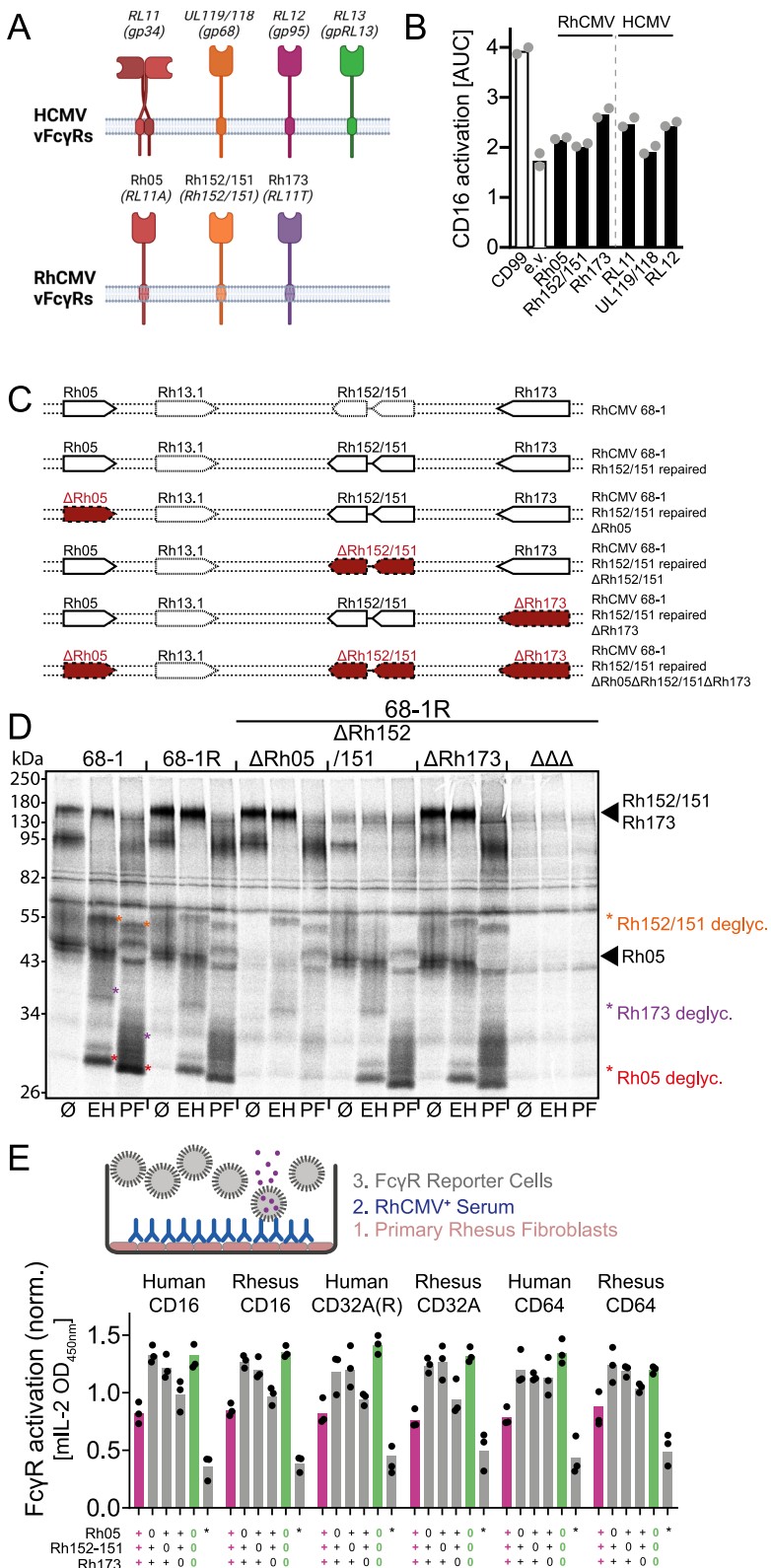

presented here raise the intriguing possibility that vFcγRs play a role in limiting the activity of antibody therapies. Blocking vFcγR function, e.g., by antibody Fab fragments that prevent Fcγ-binding, could thus improve the ability of passively administered monoclonal or polyclonal antibodies to control viral replication in immunocompromised individuals, and possibly interrupt transmission to the fetus or ameliorate fetal disease. This hypothesis is supported by recent studies demonstrating that not antibody-mediated neutralization of infection, but rather efficient FcγR-mediated effector mechanisms are correlated to protection from cross-placental CMV transmission[19,20]. Further, the inclusion of vFcγRs as antigens in an HCMV vaccine candidate may generate antibody responses that could block vFcγR-based inhibition of protective vaccine-elicited humoral immunity. Given our established RM model of congenital CMV transmission[12], we now have the unique opportunity to test these hypotheses in a relevant animal model for human vaccine development.

**Fig. 4 | RhCMV vFcγRs mirror HCMV vFcγR function and efficiently antagonize host FcγR activation in vitro. A** Graphical overview of HCMV vFcγRs and their RhCMV orthologs. Created in BioRender. Kolb, P. (2024) https://BioRender.com/f19l834. **B** Transfected HeLa cells expressing both the rhesus CD4 target antigen and the indicated vFcγRs as T2A-linked fusion protein from a pIRES-e-GFP plasmid were incubated with graded amounts of a rhesus CD4-specific rhesusized IgG1 antibody and tested for human CD16 activation using a cell-based reporter assay. Equal transfection was monitored via polycistronic GFP expression. Empty vector (e.v.) transfection and expression of a non-Fcγ binding glycoprotein (CD99) served as controls. Symbols show mean area under curve (AUC) values of independent experiments. Bars show the mean of independent experiments. **C** Overview of the RhCMV 68-1 based recombinants. To create RhCMV 68-1 R, we repaired a premature termination codon in *Rh152/151*. Single and triple deletion mutants of the vFcγRs were based on the repaired 68-1 strain. All deletions are indicated by highlighting the corresponding ORFs in red. **D** Telomerized rhesus fibroblasts (tRF) were infected with RhCMV 68-1 or 68-1 R or 68-1 R derived single vFcγR deletion

mutants at an MOI of 5 for 72 h and then labeled with [$^{35}$S]-Met/Cys for 2 h. Whole-cell lysates were prepared, and precipitations of IgG binding molecules were performed using ProteinG bound rhesus-CD4 specific rhesusized IgG1. Lysates from each condition were split and either left untreated or digested with either EndoH or PNGaseF. All samples were analyzed via gradient SDS-PAGE and subsequent autoradiography. One of two independent experiments is presented here. **E** RFs were infected with RhCMV 68-1, 68-1 R, or single vFcγR deletion mutants at an MOI of 2 for 48 h and then incubated with a 1:10 dilution of a pooled sera from eight CMV seropositive RM. The samples were assessed for human or rhesus FcγR activation using a cell-based reporter assay. The resulting mIL-2 ELISA OD values were normalized to the mean activation on the vFcγR deleted virus for each subgroup. vFcγRs expressed by viruses used for infection are indicated below. FL-RhCMV infection is indicated in pink, and complete vFcγR deletion is indicated in green. * = mock-infected cells. Symbols show independent experiments performed in technical replicates. Bars show the mean of independent experiments. Source data are provided as a Source Data file.

## Methods

### Cells

All cells were cultured in a 5% CO$_2$ atmosphere at 37 °C. Telomerized rhesus fibroblasts (tRF), HEK293T cells, 293T-CD20 (kindly provided by Irvin Chen, UCLA[55]) and HeLa cells were maintained in Dulbecco's modified Eagle's medium (DMEM, Gibco) supplemented with 10% (vol/vol) fetal calf serum (FCS, Biochrom) and antibiotics (1x Pen/Strep, Gibco). tRF were generated from primary rhesus fibroblasts (RF) obtained from animals housed at the Oregon National Primate Research Center (ONPRC) and life-extended as described previously[56]. BW5147 mouse thymoma cells (kindly provided by Ofer Mandelboim, Hadassah Hospital, Jerusalem, Israel) were maintained at $3 \times 10^5$ to $9 \times 10^5$ cells/ml in Roswell Park Memorial Institute medium (RPMI GlutaMAX, Gibco) supplemented with 10% (vol/vol) FCS, antibiotics, sodium pyruvate (1x, Gibco) and β-mercaptoethanol (0.1 mM, Gibco). Rhesus retinal pigment epithelial (RPE) cells were a kind gift from Dr. Thomas Shenk (Princeton University, USA) and were cultured in a mixture of DMEM and Ham's F12 at a 1:1 ratio supplemented with 5% FBS, antibiotics (1 × Pen/Strep, Gibco), 1 mM sodium pyruvate, and nonessential amino acids.

### Cloning of RhCMV encoded vFcγRs and vFcγR candidates into expression plasmids

Sequences of RhCMV vFcγRs were synthesized as gBlock fragments (IDT) and cloned via PstI and BamHI digest into a p-IRES-eGFP vector (Addgene) providing a unified tapasin signal peptide resulting in an N-terminal HA-epitope tag upon cleavage of the tapsin signal peptide[57]. Original signal peptides were predicted using SignalP 5.0 (DTU). CD4-tailed constructs in which the C-terminal TM and cytoplasmic domains were replaced with those of human CD4, were designed as described previously[34] and synthesized as gBlocks. A self-cleaving T2A peptide sequence was introduced into the 3′ end of the RhCMV ORF replacing the stop codon as described elsewhere[37]. gBlock fragments were designed to encode Rhesus-CD4 (GenBank: M31134.1) and fused to the individual ORFs via the T2A peptide by insertion downstream of the ORFs cloned into p-IRES-eGFP.

### Immunoprecipitation of purified Fab, Fc and IgG from metabolically labeled cells

tRFs were seeded at $1 \times 10^6$ cells per well (6-well plate) and infected at a multiplicity of infection (MOI) of 5 with the RhCMV strains 68-1, 68-1 R, or mutants thereof. Cells were labeled with [$^{35}$S]-Met/Cys for 2 h at 72 h post-infection (hpi). After NP-40 cell lysis, immunoprecipitation of proteins was performed by incubation of lysates with 10 μg/ml anti-Rhesus-CD4 IgG1 (Nonhuman Primate Reagent Resource Cat#PR-0407) for 1 h followed by retrieval of proteins using protein A Sepharose beads (PAS). Proteins were selectively deglycosylated using Endo H or PNGasF digestion and subsequently separated on a 10% to

13% SDS-PAGE gradient. Metabolic labeling was visualized using a Typhoon phosphoimager (Cytiva).

### Parental virus strains and construction of recombinants

All RhCMV recombinants generated for this study were based on bacterial artificial chromosomes (BAC) of either the tissue culture adapted RhCMV 68-1 strain (GenBank #MT157325) or full-length (FL)-RhCMV (GenBank #MT157327) which was constructed by reversing genomic inversions, deletions and ORF truncations identified in 68-1[2]. The FL-RhCMV clone now contains a complete genome representative of primary, low-passage viral isolates and demonstrates enhanced virulence and replication in immunocompetent RM when compared to 68-1[2]. FL-RhCMV-based recombinants either contained riboswitches introduced into the 5′ and the 3′ flanking regions of the Rh13.1 ORF[2], or replacements of the entire Rh13.1 ORF with heterologous antigens derived from simian immunodeficiency virus (SIV). Since intact Rh13.1 impairs growth in vitro, similar to HCMV RL13[58,59], these modifications allow for high-titer stock production without selection against the Rh13.1 ORF[2]. In addition, we used the inserted antigens as immunological makers after inoculating CMV-positive animals since we can monitor the development of antigen-specific T-cell responses as an indicator for infection and re-infection.

FL-RhCMVΔRh13.1/SIVgag/Rh05-HA/Rh173-V5, FL-RhCMVΔRh13.1/SIVgag/ΔRh05/ΔRh152/152ΔRh173, and FL-RhCMVΔRh13.1/SIV5′pol/ΔRh05/ΔRh152/152ΔRh173 were all based on RhCMVΔRh13.1/SIVgag, which has been described before[42]. HA and V5 were added to Rh05 and Rh173 through sequential homologous *en passant* recombination using specific primer pairs containing the DNA sequence for the peptide tags. The triple vFcγR deletion mutant (ΔΔΔ) was based on FL-RhCMVΔRh13.1/SIVgag and constructed by sequentially deleting the entire coding region of each ORF. For superinfection experiments using the ΔΔΔ recombinant in RhCMV seropositive RM, SIVgag was replaced with SIV5′pol. RhCMV 68-1 R was constructed by repairing the SNP resulting in a premature termination codon in Rh152/Rh151 by homologous *en passant* recombination. The same techniques was then used to either create single deletion mutants of individual vFcγR (Rh05, Rh152/151, or Rh173) or a triple (ΔΔΔ) vFcγR deletion mutants in which all ORFs were sequentially fully deleted.

The *en passant* recombination technique[60] used to introduce alterations into the parental BACs had been previously adapted to our RhCMV vector system[42]. In short, this technique requires the duplication of a 50-bp stretch of the genome flanking the targeted genomic locus. This duplication can be achieved through strategic primer design. These primers are then used to amplify an aminoglycoside 3-phosphotransferase gene conferring kanamycin resistance (KanR) as a selection marker preceded by an I-SceI homing enzyme target sequence. After homologous recombination, the selection marker is flanked by the introduced direct repeats. Expression of I-Sce I by

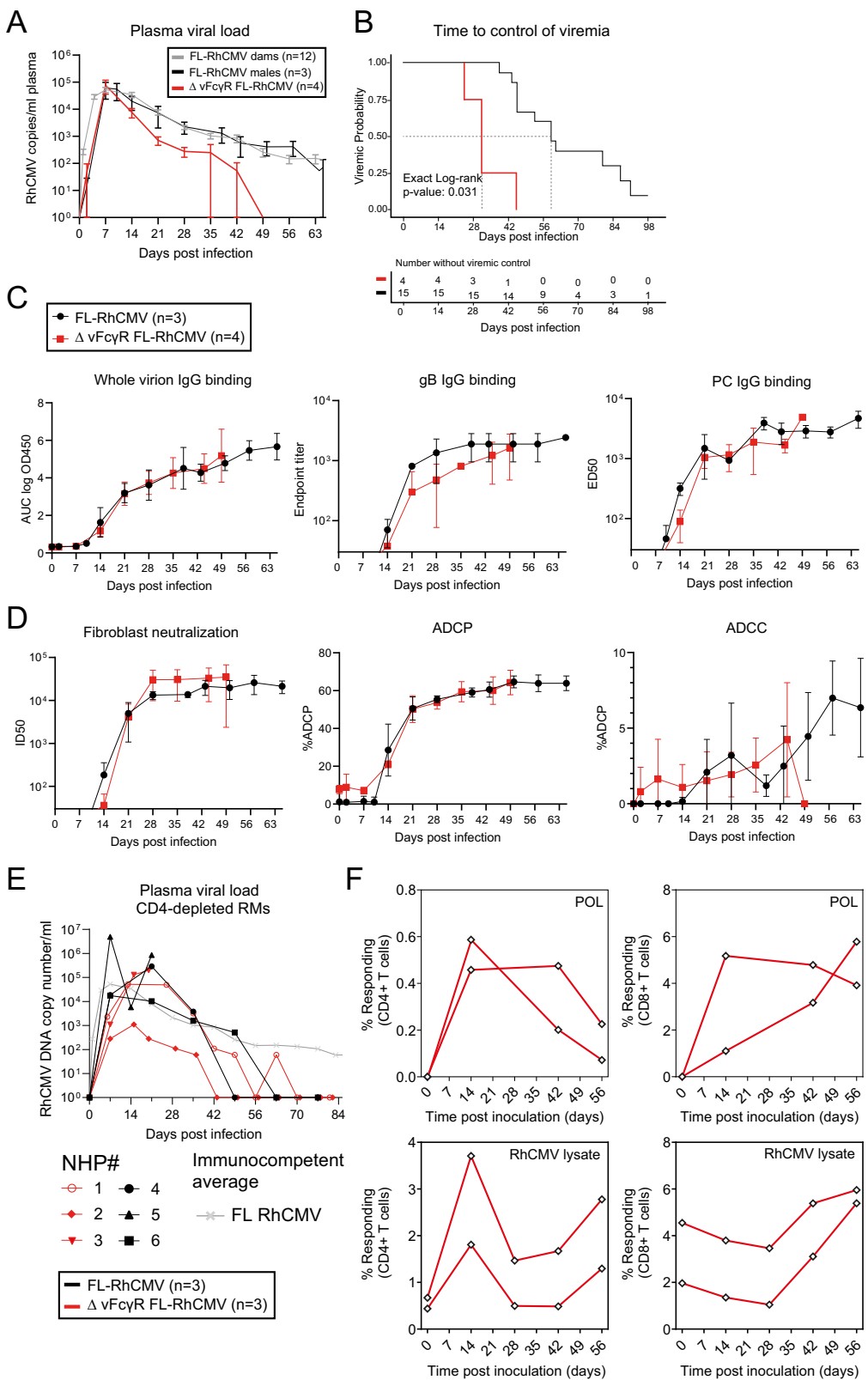

arabinose induction in *E.coli* strain GS1783 harboring the RhCMV BACs results in the selective induction of DNA double-strand breaks in the RhCMV BAC. These breaks are then repaired via recombination of the repeated DNA sequences by inducing the expression of the λ phage-derived Red recombination genes through heat shock. This results in the complete removal of the KanR resistance cassette without retaining any DNA sequences introduced during recombineering. Final

constructs were analyzed by Xma I restriction digest, Sanger sequencing of the altered genome region, and full-genome analysis by next-generation sequencing (NGS) before virus reconstitution.

## Multistep growth curves

Primary rhesus fibroblasts (RF) were seeded at $5 \times 10^4$ cells per well in 24-well plates (Corning). The cells were infected in triplicate with

**Fig. 5 | RhCMV encoded vFcγRs antagonize the clearance of viral genomes from plasma in vivo. A** Four RhCMV-seronegative male RM were inoculated with an FL-RhCMVΔΔΔ. Three RhCMV-seronegative male RM and 12 RhCMV-seronegative female RM were inoculated with FL-RhCMV or FL-RhCMV plus RhCMV UCD52 respectively. DNA was isolated form plasma samples, and viral genome copy numbers were determined by qPCR using primer/probe sets targeting exon 1 of the IE locus. Copy numbers are reported as mean +/− SD via standard curve interpolation. **B** Kaplan-Meyer survival analysis showing time to control of plasma DNAemia (first-time point with genome copy numbers below the detection limit (1 copy/well). The median time to control was 31.5 days post-infection in animals infected with vFcγR-deleted RhCMV compared to 59.5 days in animals infected with FL-RhCMV ($p = 0.031$). **C** IgG binding to whole FL-RhCMV virions, RhCMV gB, and RhCMV pentamer was measured by ELISA and is shown as mean +/− SD. **D** ADCP is reported as the percentage of live THP-1 cells containing fluorescently labeled FL-RhCMV after incubation with plasma. ADCC is reported as the percentage of live rhesus CD16-expressing NK92 cells expressing the degranulation marker CD107a following co-incubation with FL-RhCMV infected fibroblasts and plasma samples. All responses are shown as mean +/− SD and demonstrate comparable magnitude and kinetics between animals infected with FL-RhCMV ($n = 3$) versus vFcγR-deleted RhCMVΔΔΔ ($n = 4$). **E** Kinetics of viral genome copy numbers in the plasma of CD4 + T cell-depleted RhCMV-naïve (seronegative) animals infected with vFcγR-deleted RhCMV (red) versus FL-RhCMV (black). Averages from immunocompetent RhCMV-seronegative RM infection in (A) are shown for comparison (gray). **F** Two RhCMV-seropositive RMs were inoculated with $5 \times 10^6$ PFU of vFcγR-deleted RhCMVΔΔΔ carrying an SIV-5'pol transgene as an immunological marker replacing the Rh13.1 ORF. The onset of SIV 5'Pol-specific CD4+ and CD8 + T cell responses and the boosting of RhCMV-specific T cell responses were measured in peripheral blood by intracellular cytokine staining (ICS) for IFN-γ and TNF-α using either a pool of 15mer peptides overlapping by 11 amino acids (AA) corresponding to SIV-5'pol[52] or RhCMV lysates. The frequency of IFN-γ + and/or TNF-α + memory T cells is shown for each individual animal, and each indicated time point post-inoculation. Gating Strategies in Fig. S10. Source data are provided as a Source Data file.

either FL-RhCMV ΔRh13.1/SIVgag or the FL-RhCMV ΔRh13.1/SIVgag ΔRh05ΔRh152/Rh151ΔRh173 (ΔΔΔ) recombinant at a multiplicity of infection (MOI) of 0.01. Starting from day 3 post-infection, supernatants were collected every third day until day 24 and stored at < 80 °C. To determine the number of focus forming units (FFU) in each collected sample, $10^4$ RF per well were seeded into 96-well plates (Corning) in quadruplicates and infected the next day with the supernatant samples (1:5 serial dilution). At 72 hpi, the cells were fixed with 100% methanol for 20 min at − 20 °C and then permeabilized with 2% paraformaldehyde and 0.1% Triton X-100 in Phosphate Buffered Saline (PBS) for 15 min at room temperature (RT). Subsequently, the cells were blocked for 30 min with 2% BSA in PBS (PBA) and stained with a mouse anti-RhCMV pp65b mAb (clone 19C12.2, generated in house by OHSU monoclonal antibody shared resource[61]) for 1 h at 37 °C. After washing three times with PBA, the cells were stained with an Alexa488-conjugated anti-mouse secondary antibody (Invitrogen) for 1 h at 37 °C and then with 3 μM DAPI (Invitrogen) for 20 min. Images were acquired using an EVOS fluorescence microscope (Life Technologies), and the ImageJ software was used to process the images and count all pp65b + and DAPI + cells. Double positive versus single positive (DAPI +) cell counts were used to calculate the virus titer for each sample as focus forming units per milliliter (FFU/ml).

## Quantitative reverse transcriptase PCR (qRT-PCR) analysis to determine the kinetics of RhCMV FcγRs transcripts
RF were seeded in 6-well plates and infected with either FL-RhCMV or FL-RhCMV ΔRh05ΔRh152/151ΔRh173 (both also containing ΔRh13.1/SIVgag) as a negative control at a MOI of 5. Subsequently, total RNA was isolated using TRIzol reagent (Thermo Fisher Scientific) at 4, 8, 12, 24, 36, and 48 hpi following the manufacturer's instructions. Next, 1 μg of total RNA per sample was DNase I (Zymo Research) treated and then used to synthesize complementary DNA (cDNA) using the Maxima Reverse Transcriptase (Thermo Fisher Scientific). To determine the transcription levels and temporal kinetics of the RhCMV FcγRs *Rh05, Rh152/151, and Rh173*, qPCR was performed with the cDNA using TaqMan Fast Advanced Master Mix (Applied Biosystems) and QuantStudio 7 Flex Real-Time PCR Systems (Applied Biosystems). Transcript copy numbers of target genes were determined using the QuantStudio Real-Time PCR Software v1.3 and then normalized to the housekeeping transcript for Glyceraldehyde 3-phosphate dehydrogenase (GAPDH). Data were graphed for each gene and time point as relative mRNA copy numbers. To determine the kinetic class of each vFcγR transcript, the immediate early (IE) gene *Rh156*, the early (E) gene *Rh189*, and the late (L) gene *Rh137* were included as controls as described previously[42]. For the qPCR assay, primers and probes specific to each gene of interest were used (Supplementary Table S1). Expression of genes adjacent to the deleted genes in FL-RhCMV ΔRh05ΔRh152/151ΔRh173 was determined by qualitative PCR using the same cDNA and primer sets specific to each deleted gene and the corresponding adjacent genes (Supplementary Table S1). To confirm the absence of residual viral DNA in the cDNA samples, a set of RNA samples was used during cDNA synthesis without reverse transcriptase (RT-) treatment.

## RhCMV virion isolation and virion proteins detection by immunoblot
FL-RhCMVΔRh13.1/SIVgag, FL-RhCMVΔRh13.1/SIVgag/ΔRh05/ΔRh152/ 152ΔRh173 (ΔΔΔ) and FL-RhCMVΔRh13.1/SIVgag/Rh05-HA/Rh173-V5 virions were purified over a discontinuous Nycodenz gradient, as described previously[61]. The virus was isolated from the culture medium of infected RF. For each virus, 10 T-175 flasks of confluent RFs were infected at a MOI of 0.05-0.1. Infected cells were harvested after all cells showed maximal cytopathic effect. The supernatants were first clarified by centrifugation at $7500 \times g$ for 15 min. The clarified medium was layered over a sorbitol cushion (20% D-sorbitol, 50 mM Tris [pH 7.4], 1 mM MgCl$_2$), and virus was pelleted by centrifugation at $64,000 \times g$ for 1 h at 4 °C in a Beckman SW28 rotor. The virus pellet was resuspended in TNE buffer (50 mM Tris [pH 7.4], 100 mM NaCl, and 10 mM EDTA). The virus particles were further purified by layering them over a discontinuous 5% to 50% Nycodenz gradient (Sigma-Aldrich) in TNE buffer, followed by centrifugation at $111,000 \times g$ for 2 h at 4 °C in a Beckman SW41 Ti rotor. The virion bands in the gradient were extracted with a syringe through the side of the centrifuge tube, and the particles were pelleted in a Beckman TLA-45 rotor in a Beckman Optima TL 100 Ultracentrifuge at $40,000 \times g$ for 1 h and washed twice with TNE buffer. In order to detect RhCMV FcγRs in the RhCMV virion preparation, pellets were lysed with 200 μl RIPA lysis buffer (Thermo Fisher Scientific) and 200 μl of 2x Laemmli buffer (0.125 M Tris HCl pH 6.8, 4% SDS, 20% glycerol, 0.004% bromophenol blue). Protein lysates were denatured at 95 °C for 10 min and similar protein amounts were electrophoretically separated on a NuPAGE™ 10%, Bis-Tris gel (Invitrogen, Thermo Fisher Scientific), electrophoretically transferred onto PVDF transfer membranes (Thermo Fisher Scientific) and stained for HA (Sigma-Aldrich, HA-7), V5 (Invitrogen, E10/V4RR), Rh152/151, and Rhpp65b using specific antibodies. Rh152/151 and Rhpp65b were made by OHSU monoclonal antibody shared resource[43]. Protein bands were visualized by SuperSignal™ West pico PLUS chemiluminescent substrate (Thermo Fisher Scientific).

## Comparison of fibroblasts and epithelial cell infections
RF and rhesus retinal pigment epithelial cells (RPE) were seeded at 12,000 cells per well onto 96-well plates. After 24 h, the cells were infected with either FL-RhCMV, vFcγRs deleted FL-RhCMV (ΔΔΔ), or pentameric complex (PC)-deleted FL-RhCMV (deleted for Rh157.5 and Rh157.4, the homologs of HCMV UL128 and UL130, all ΔRh13.1/SIVgag) at a 1:3 serial dilution starting at 1:50 inoculum in DMEM complete. At 72 hpi, all cells were fixed with methanol for 20 min at − 20 °C, then

washed three times with PBS. Subsequently, fixed cells were stained with the mouse α-RhCMV pp65b antibody (clone 19C12.2, generated in house by OHSU monoclonal antibody shared resource) for 1 h at 37 °C. Next, cells were stained with Alexa488-conjugated anti-mouse secondary antibody (Thermo Fisher Scientific) for 1 h at 37 °C and then with DAPI (Thermo Fisher Scientific) for 10 min at RT. Images were acquired using the EVOS® FL Auto Imaging System (Thermo Fisher Scientific) and analyzed using the ImageJ software. Relative infection rates were calculated as the ratio of focus forming units per ml (FFU/ml) in RPE cells versus FFU/ml in RF for each recombinant. Experiments were performed in triplicate in three biological repeats.

### Immunofluorescence assay and fluorescence microscopy

tRFs were cultured in DMEM supplemented with 10% FBS, 100 μ/ml penicillin, and 100 μg/ml streptomycin. $10^6$ TRFs were plated overnight on 12 mm wide, 0.13–0.17 mm thick glass coverslips in 24-well plates. The next day, cells were infected with FL-RhCMV, FL-RhCMVΔRh152/152, or FL-RhCMV/Rh05-HA/Rh173-V5 at MOI of 1. At 24 hpi and 48 hpi, cells were rinsed with PBS and fixed with 4% methanol-free formaldehyde for 10 min at RT. Cells were washed three times using PBS and blocked and permeabilized using saponin-BSA buffer for 1 h at RT (0.2% saponin from Millipore and 1% BSA from Fisher). Cells were incubated with the following primary antibodies for 1 h at RT: Anti-HA (Abcam-Ab18181), Anti-RhCMV-Rh152/151 (mouse mAb generated in house by OHSU monoclonal antibody shared resource[43]). Anti-V5 (GeneTex-GTX628529), Anti-RhCMV IE2 (mouse mAb generated in house by OHSU monoclonal antibody shared resource[43]), Anti-EEA1 (CST-2411S), Anti-LAMP1 (Thermo Fisher Scientific, PA1-654A), Anti-Rab6A (Thermo Fisher Scientific, PA5-22127), Anti-ERp57 (Invitrogen-PA3-009). Coverslips were washed three times for a total of 30 min using saponin-BSA buffer. Secondary antibodies (A21463 anti-mouse 647, A21441 anti-rabbit 488, A21155 IgG3-594, A21131 IgG2a 488, A21240 IgG1 647 (all from Thermo Fisher Scientific) and DAPI (D1306, Thermo Fisher Scientific) in saponin-BSA buffer were incubated for 1 h at RT. Cells were washed thrice for a total of 30 min in saponin-BSA buffer followed by a final wash using PBS. Coverslips were mounted on a glass slide using Prolong Gold (P36930, Thermo Fisher Scientific) and cured for 24–48 hours. The cells were imaged using the 100x oil immersion objective of a Keyence BZ-X710 microscope using its built-in 2.8 megapixel monochrome CCD camera. To eliminate fluorescence blurring in images, the sectional function of 2D structured illumination and haze reduction built-in features were used. Data was processed using FIJI software. All the immunofluorescence experiments were independently repeated four times.

### Flow cytometry on transfected cells

Cells were washed in PBS, equilibrated in staining buffer (PBS, 3% FCS), and pelleted at $1000 \times g$ and 10 °C for 3 min. Cells were incubated in a staining solution (antibody diluted in the buffer as suggested by the supplier). Every incubation step was carried out at 4 °C for 1 h, followed by 3 washing steps in staining buffer. Dead cells were stained using DAPI. Cells were analyzed on a FACS Fortessa LSR instrument (BD Bioscience). Intracellular stains were performed using the Cytofix/Cytoperm kit according to the supplier's instructions (Becton Dickinson). HA-epitopes were detected using PE-labeled clone 16B12 (Biolegend). PE-conjugation was performed using an ab102918 labeling kit (Abcam) as suggested by the supplier. Data was evaluated using FlowJo 10 software. Gating was used to identify transfected cells (GFP expression from pIRES_eGFP vector) from all cells only.

### Fcγ-receptor activation assay

The assay was performed as described previously[28,34,39]. Briefly, target cells were incubated with dilutions of a rhesus-CD4 specific rhesusized IgG1[28] or a serum pool from 8 RhCMV-seropositive RM in DMEM supplemented with 10% (vol/vol) FCS for 30 min at 37 °C. Cells were washed

before co-cultivation with FcγR-activation reporter cells (ratio E:T 10:1) for 16 h at 37 °C in a 5% $CO_2$ atmosphere. For all activation assays, mouse IL-2 secretion was quantified by anti-IL-2 ELISA as described previously[39].

### CD16 binding assay

The assay was performed as described previously[34]. In brief, 293T-CD20 cells expressing transfected vFcγRs or control constructs were harvested using Accutase® (Sigma-Aldrich) to retain surface molecules upon detachment. Harvested cells were washed in PBS, equilibrated in staining buffer (PBS, 3% FCS), and sedimented at $1000 \times g$ and 10 °C for 3 min. Cells were then incubated with staining buffer containing a humanized human CD20-specific mAb (rituximab) followed by incubation with recombinant human or rhesus FcγR ectodomains pre-incubated with a His-tag-specific PE-labeled mAb. His-tagged FcγR (human CD16A/FcγRIIIA F176V) ectodomains were used at a final concentration of 5 μg/ml (10389-H08H1 and 90013-C27H, Sino Biological). Dead cells were labeled via DAPI stain. Analysis was performed on a FACS Fortessa LSR instrument (BD Bioscience).

### Rhesus macaques

At the Tulane Primate National Research Center (TNPRC) RhCMV-seronegative expanded specific pathogen–free (eSPF) RMs of Indian-origin RMs were housed in accordance with institutional and federal guidelines for the care and use of laboratory animals[62]. Protocols were approved by the TNPRC Institutional Animal Care and Use Committee (IACUC) and conducted in accordance with the Guide for the Care and Use of Laboratory Animals at the NIH. All animals were screened for RhCMV-specific IgM and IgG by whole virion ELISA and confirmed RhCMV-seronegative before being enrolled in this study. Assigned male RM were inoculated i.v. with $1 \times 10^6$ plaque-forming units (PFU) of either FL-RhCMV ($n = 3$) or vFcγR-deleted FL-RhCMVΔΔΔ ($n = 4$) while a cohort of pregnant, eSPF dams ($n = 12$) was inoculated with $1 \times 10^6$ PFU of FL-RhCMV and UCD52 each in the late first/early second trimester of pregnancy. Blood samples were collected at 0, 2, 7, 10, 14, 21, 28, 38, 43, 50, 57, and 64 or 65 days post-infection (dpi) for FL-RhCMV infected animals and 0, 2, and 7 dpi and weekly thereafter through 6-7 weeks post-infection for FL-RhCMVΔΔΔ infected animals as well as the cohort of pregnant dams.

At the California National Primate Research Center (CNPRC), protocols were approved prior to implementation by the Institutional Animal Care and Use Committee (IACUC). Activities related to animal care, housing, and diet were performed according to CNPRC standard operating procedures (SOPs). Normally cycling, adult female RM confirmed seronegative for RhCMV and with a history of prior pregnancy ($n = 4$; one in the FL-RhCMV group [#3] and three in the vFcγR-deleted group [#4, #5, #6]) were time-mated according to established methods, with pregnancy identified by ultrasound early in gestation[63]. Two gravid animals in the FL-RhCMV group (#1 and #2) were assigned to the study after pregnancy had been achieved spontaneously in a group housing environment. They were relocated to indoor housing prior to study initiation. All fetuses were sonographically assessed to confirm normal growth and development prior to maternal interventions (e.g., CD4 depletion, RhCMV inoculation). The dams were administered ketamine hydrochloride (10 mg/kg, intramuscular-IM) or telazol (5–8 mg/kg; IM) for these and subsequent ultrasound examinations. Prior to virus inoculation, all dams received recombinant rhesus anti-CD4 antibody (CD4R1 clone at 50 mg/kg IV; provided by the Nonhuman Primate Reagent Resource) to achieve CD4 + T cell depletion, which was verified by flow cytometry (compared to the baseline sample collected, Supplementary Fig. S9). Animals were then inoculated i.v. with $1 \times 10^6$ PFU of FL-RhCMV and UCD52 each (total injection volume 1 ml) ($n = 3$) or $1 \times 10^6$ PFU of vFcγR-deleted FL-RhCMVΔΔΔ ($n = 3$) in the late first/early second trimester of pregnancy. Maternal blood samples were collected weekly throughout the study period (prior to CD4 antibody depletion then weekly).

At the Oregon National Primate Research Center (ONPRC) the two assigned male RMs of Indian origin were housed as a pair with visual, auditory, and olfactory contact with other animals in an Animal Biosafety level (ABSL)-2 room with autonomously controlled temperature, humidity, and lighting. Protocols were approved by the ONPRC and the TNPCR Institutional Animal Care and Use Committees (IACUC). Both institutions are Category I facilities, and they are fully accredited by the Assessment and Accreditation of Laboratory Animal Care International and have an approved Assurance (#A3304-01) for the care and use of animals on file with the NIH Office for Protection from Research Risks. The IACUCs adhere to national guidelines established in the Animal Welfare Act (7 U.S.C. Sections 2131–2159) and the Guide for the Care and Use of Laboratory Animals (8th Edition) as mandated by the U.S. Public Health Service Policy. The animals were inoculated with $5 \times 10^6$ PFU of a vFcγR-deleted RhCMVΔΔΔ carrying an SIV-5'pol transgene as an immunological marker replacing the Rh13.1 ORF, and blood samples were collected every other week starting a day 0 post-infection. The RM were administered ketamine (~7 mg/kg) prior to blood collection from the femoral or saphenous veins while the samples were collected using Vacutainers. The animals were humanely euthanized by the veterinary staff at ONPRC in accordance with end-point policies and as recommended by the American Veterinary Medical Association.

## Viral load qPCR
DNA was isolated using the QIAmp DNA minikit (Qiagen). We utilized a 40-cycle real-time qPCR reaction using 5'-GTTTAGGGAACCGC-CATTCTG-3' forward primer, 5'-GTATCCGCGTTCCAATGCA-3' reverse primer, and 5'-FAM-TCCAGCCTCCATAGCCGGGAAGG-TAMRA-3' probe directed against a 108 bp region of the highly conserved RhCMV *IE* gene, using a standard IE plasmid to interpolate the number of viral DNA copies/ml of plasma. Two out of six replicates above the limit of detection (1–10 copies per well) were required to report a positive result for each plasma sample.

## Antibody serology assays
RhCMV-specific IgG antibody kinetics were measured in plasma by enzyme-linked immunosorbent assays (ELISAs) using either whole virion preparations of FL-RhCMV or purified glycoprotein preparations of gB or pentamer as described previously[13]. Briefly, high-binding 384-well ELISA plates (Corning) were coated with 15 μl/well of 5120 pfu/ml RhCMV or 2 μg/ml purified gB or pentameric complex (PC) protein in 0.1 M sodium bicarbonate (pH = 9.55) overnight at 4 °C. Following overnight incubation, plates were blocked for 1-2 h with blocking solution (PBS +, 4% whey protein, 15% goat serum, 0.5% Tween-20), and then 3-fold serial dilutions of plasma (1:30 to 1:65,610) were added to the wells in duplicate for 1-2 h. Plates were then washed using an automated plate washer (BioTek) and incubated for 1 h with mouse anti-rhesus IgG HRP-conjugated secondary antibody (Southern Biotech, clone SB108a) at a 1:5000 dilution or anti-rhesus IgM HRP-conjugated secondary antibody (Rockland) at 1:8000. After two washes, SureBlue Reserve TMB Microwell Peroxidase Substrate (KPL) was added to the wells for 7 min for whole virion ELISAs and 3.5 min for glycoprotein ELISAs. The reaction was stopped by the addition of 1% HCl solution (KPL), and plates were read at 450 nm. The cutoff for positive antibody reactivity was 3 standard deviations (SDs) above the average $OD_{450}$ measured for RhCMV-seronegative samples at the starting plasma dilution (1:30). $ED_{50}$ was calculated as the sample dilution where 50% binding occurs by interpolation of the sigmoidal binding curve using four-parameter logistic regression, and this value was set to half of the starting dilution (15) when the 1:30 dilution point was negative by our cutoff and when the calculated $ED_{50}$ was below 15. Endpoint titer (AUC) was used to report binding results when the dilution series resulted in incomplete sigmoidal curves and, thus, unreliable $ED_{50}$ calculations for the majority of positive samples. The

endpoint titer is defined as the last dilution factor at which there was binding above the positivity cutoff. AUC was calculated by integrating the $OD_{450}$ over the full dilution series.

Neutralization of RhCMV on fibroblasts was monitored as previously described[13]. Briefly, serial dilutions (1:30 to 1:65,610) of heat-inactivated rhesus plasma were incubated with FL-RhCMV[2] (MOI = 1) for 1–2 h at 37 °C. The virus/plasma dilutions were then added in duplicate to 384-well plates containing confluent cultures of TRF cells and incubated at 37 °C for 24 h. Infected cells were then fixed for 20 min in 10% formalin and processed for immunofluorescence with 1 μg/ml mouse anti-RhCMV pp65B or 5 μg/ml mouse anti-Rh152/151 monoclonal antibody followed by a 1:500 dilution of goat anti-mouse IgG-Alexa Fluor 488 antibody (Invitrogen). Nuclei were stained with DAPI for 10 min (Invitrogen). Infection was quantified in each well by automated cell counting software using a Molecular Devices ImageX-press Pico. Subsequently, the $ID_{50}$ was calculated as the sample dilution that caused a 50% reduction in the number of infected cells compared with wells treated with virus only. If the $ID_{50}$ was below the limit of detection (e.g., every value in the series resulted in a level of infection above 50%), we set the $ID_{50}$ to 15 (half of the first dilution).

Antibody-dependent cellular phagocytosis was assessed by conjugation of concentrated FL-RhCMV to Alexa Fluor 647 using NHS-ester reaction (Invitrogen), which was allowed to proceed in the dark with constant agitation for 1–2 h and then quenched by the addition of pH 8.0 Tris hydrochloric acid, and 5000 pfu of the conjugated virus (10 μl) of virus and plasma samples in duplicate at a 1:30 dilution were combined in a 96-well U-bottom plate (Corning) and incubated at 37 °C for 2 h. THP-1 monocytes were then added at 50,000 cells per well. Plates were spun for 1 h at $1200 \times g$ at 4 °C and then transferred to a 37 °C incubator for an additional 1 h. Cells were then washed with 1% FBS in PBS and stained with aqua live/dead (Invitrogen) at 1:1000 for 20 min. Following another wash step, cells were fixed for 20 min in 10% formalin and resuspended in PBS. Fluorescence was measured using a BD Fortessa flow cytometer using the high-throughput sampler (HTS) attachment, and data analysis was performed using FlowJo software (v10.8.1). Results are reported as the percent of the live population that was AF647 +.

Antibody-dependent cellular cytotoxicity was measured by NK cell degranulation using NK92.rh.158I.Bb11, an engineered NK cell line expressing the Ile158 variant of rhesus macaque CD16[64]. Confluent TRF cells were infected with 1 MOI FL-RhCMV in low serum (5%) medium in T75 flasks (Thermo Fisher Scientific). Mock infection was performed in parallel. After 24 h of infection, cells were dissociated from the flask using TrypLE (Gibco), and 50,000 target cells were seeded in each well of a 96-well flat bottom tissue culture plate (Corning). Cells were incubated for 16–20 h to allow them to adhere. Then, the target cell medium was removed, and 50,000 NK92.Rh.CD16-Bb11 cells were added to each well with Brefeldin A (GolgiPlug, 1:1000, BD Biosciences), monensin (GolgiStop, 1:1500, BD Biosciences), CD107a-FITC (BD Biosciences, 1:40, clone H4A3), and plasma samples at 1:25 dilution or purified IgG controls from either seropositive or seronegative plasma donors at 50–250 μg/ml in RPMI1640 containing 10% FBS, plated in duplicate, and incubated for 6 h at 37 °C and 5% $CO_2$. The same samples were added to mock-infected target cells for background subtraction. NK cells were then collected and transferred to a 96-well V-bottom plate (Corning). Cells were pelleted and resuspended in aqua live/dead diluted 1:1000 for a 20 min incubation at RT. Cells were washed with PBS + 1% FBS and stained with CD56-PECy7 (BD Biosciences, clone NCAM16.2) and CD16 PacBlue (BD Biosciences, clone 3G8) for a 20 min incubation at RT. Cells were washed twice and then resuspended in 10% formalin for 20 min. Cells were then resuspended in PBS for acquisition. Events were acquired on a BD Symphony A5 using the HTS attachment, and data analysis was performed using FlowJo software (v10.8.1). Data is reported as the % of CD107a + live NK cells (singlets, lymphocytes, aqua blue–, CD56 +, CD107a +) for each sample. The background was subtracted according to antibody

control wells within mock and infected conditions, followed by subtraction of the mock signal.

## T cell assays

CD4 + and CD8 + T cell responses were measured in PBMC by flow cytometric intracellular cytokine staining (ICS) as described in detail previously[42]. Frozen PBMCs were rapidly thawed at 37 °C and washed with R10-media (RPMI 1640 medium, Thermo Fisher Scientific) supplemented with 10% Newborn Calf Serum (Gibco), 2 mM l-glutamine (Thermo Fisher Scientific), 100 μ/ml penicillin (Invitrogen), 100 μg/ml streptomycin (Invitrogen), 50 μM βME (Thermo Fisher Scientific) before cells were combined with anti-CD28 (CD28.2, Purified 500 ng/test: eBioscience), anti-CD49d mAb (9F10, Purified 500 ng/test: eBioscience), and test antigen then incubated for 1 h at 37 °C before the addition of Brefeldin A, followed by an additional 8 h incubation. Co-stimulation without antigen served as a negative control. Cells were then stained with the following fluorochrome conjugated antibodies: anti-CD3 (SP34−2, Pacific Blue; BD Biosciences), anti-CD8 (SK1, PerCP-eFluor710; Life Tech), anti-CD4 (L200, BV510; BD Biosciences), anti-CD69 (FN50, PE-TexasRed, BD Biosciences), anti-IFNγ (B27, APC; BD Biosciences), anti-TNFα (MAb11, PE; BD Biosciences), and Ki67 (B56; FITC, BD Biosciences). T cell responses to SIV 5'Pol were measured by ICS using a mix of sequential 15-mer peptides (11 amino acid overlap). Stained samples were analyzed on an LSR-II or FACSymphony A5 flow cytometer (BD Biosciences) and analyzed using FlowJo 10 software (BD Biosciences). In all analyses, gating on the lymphocyte population was followed by the separation of the CD3 + T cell subset and progressive gating on CD4 + and CD8 + T cell subsets. The gating strategy is shown in Supplementary Fig. S10. Antigen-responding cells in both CD4 + and CD8 + T cell populations were determined by their intracellular expression of CD69 and either or both of the cytokines IFN-γ and TNF-α. The assay limit of detection was determined with 0.05% after background subtraction being the minimum threshold used in this study. After background subtraction, the raw response frequencies above the assay limit of detection were "memory-corrected" (i.e., percent responding out of the memory population), as previously described[42].

## Statistical analysis

Survival analysis comparing time to DNAemia stabilization, defined as the midpoint between the last day where the viral load measurement was above the qPCR limit-of-detection (1 copy/well) and the first day observed after peak DNAemia where the viral load fell below limit-of-detection, between rhesus macaques infected with FL-RhCMV and FL-RhCMVΔΔ was assessed via an exact log-rank test using the coin package[65,66]. Kaplan-Meier estimates of viremic probabilities were employed for visualization and quantifying median time to DNAemia stabilization[67]. All statistical analyses were performed using R statistical software (www.r-project.org).

## Reporting summary

Further information on research design is available in the Nature Portfolio Reporting Summary linked to this article.

## Data availability

All data is available within this paper. Accession Codes (previously reported): Rhesus CD4: M31134 [https://www.ncbi.nlm.nih.gov/nuccore/M31134.1] Rhesus CMV 68-1 genome: MT157325 [https://www.ncbi.nlm.nih.gov/nuccore/MT157325] Rhesus CMV FL genome: MT157327 [https://www.ncbi.nlm.nih.gov/nuccore/MT157327]. Source data are provided in this paper.

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

## Acknowledgements

The authors would first like to acknowledge the rhesus macaques that were used in this study and thank the faculty and staff of the Departments of Veterinary Medicine and Collaborative Research at the Tulane National Primate Research Center (TNPRC) and the Oregon National Primate Research Center (ONPRC) for their excellent care of our research animals. The authors also thank the animal care and veterinary staff at the California National Primate Research Center (CNPRC) at UC Davis. We acknowledge the Molecular Virology Core at Oregon National Primate Research Center (ONPRC) for performing viral productions and qPCRs. We are additionally grateful to the NIH Nonhuman Primate Reagent Resource (R24 OD010976 and NIAID contract HHSN 272201300031 C) which provided the CD4-depleting antibody. We further wish to acknowledge support from the Biostatistics, Epidemiology, and Research Design (BERD) Methods Core. Figures 1A and 4A cartoons were generated with Biorender (biorender.com), License University Medical Center Freiburg, Institute of Virology. We extend acknowledgements to the Estes lab at the VGTI for offering their Keyence microscope for the generation of the here presented IFA data. This work was funded by the German Research Foundation (DFG KO6815/1-11 and HE2526/9-2) and awarded to P.K. and H.H. The here presented in vivo studies were funded by NIH/NIAID 3P01AI129859 and NIH/NICHD DP2HD075699 awarded to S.R.P. C.E.O. received funding from NIH/NIAID 3P01AI129859-04S1, NIH/NCI T32-CA009111, and 2TL1-TR-2386. The Tulane National Primate Research Center is supported by NIH (OD011104). The Oregon National Primate Research Center (ONPRC) and the OHSU Molecular Virology Core are supported by NIH P51OD011092. The California National Primate Research Center base operating grant (P51-OD011107), and related in vivo imaging was performed with instrumentation obtained through NIH S10 High-End Instrumentation grants to A.F.T (OD016261). The Biostatistics, Epidemiology, and Research Design Methods Core is funded by NIH/NCATS UL1TR002553. The funders had no role in the study design, data collection and interpretation, decision to publish, or the preparation of this manuscript.

## Author contributions

Conceived and designed the experiments: P.K., D.M., S.R.P., A.K., and K.F. Performed the experiments: C.E.O., M.E., S.P., H.T., N.J., K.H., P.K., M.J.M., E.A.S., L.M.S., S.K., T.D.M., N.H.V.B., A.F.T., R.G., C.P., T.B., L.K.N., M.D., and Z.S. Analyzed the data: P.K., D.M., C.E.O., A.D., A.B.A., R.B., A.F.T., D.S., and S.G.H. Writing and original draft preparation: P.K., D.M., and C.E.O. Review and editing: S.P., K.F., A.F.T., H.H., and C.C.

## Funding

## Competing interests

S.R.P. has served as a consultant to Merck, Moderna, Pfizer, GSK and Dynavax and has led sponsored programs with Moderna and Merck. P.K. has received funding from Biotest AG. None of these activities have impacted this work. The remaining authors declare no competing interests.
