## [Transparent Peer Review file · Nature Communications]

Rhesus Cytomegalovirus-encoded Fcγ-binding glycoproteins facilitate viral evasion from IgG-mediated humoral immunity

Corresponding Author: Dr Philipp Kolb

Version 0:

Reviewer comments:

Reviewer #1

(Remarks to the Author)

This study by Claire Otero and co-authors addresses the functional roles of cytomegalovirus encoded Fc binding proteins, aka "viral FcR's" during rhesus CMV (RhCMV) infection of macaques. Interestingly, HCMV, and as this study shows, RhCMV each encode multiple distinct FcR's. The literature suggests that each of the HCMV vFcRs possesses unique/ non-overlapping activities which arguably act with synergism to block antibody dependent effector functions such ADCP and ADCC. Here, Otero et al. characterize the in vivo phenotypic outcome / impact on viral fitness during natural infection when all three RhCMV vFcR are genetically ablated, finding that virus lacking all 3 RhCMV vFcR exhibits a shortened plasma viremia but that the deletion virus is otherwise able to infect previously seroconverted animals. Thus the vFcR-triple null RhCMV still is able to defy "vector immunity," a feature of CMVs that relies T cell evasion functions. The authors also provide ex vivo characterization addressing how these viral molecules dampen cellular FcR activity, and evaluating their sub cellular localization and trafficking within infected cells. The authors had previously reported that Rh05 is a vFcR, and here they start out by identifying RhCMV counterparts for two additional HCMV vFcR's that to-date had not yet been formally identified in RhCMV, at least not beyond recognition of a positional homolog ORF. They find via screening of RL11 family members that Rh173 is a vFcR. The authors also observe that Rh13.1, which sequence data suggest is the RhCMV paralog of the HCMV vFcR RL13 does not in fact exhibit IgG binding activity in their hands.

Overall this is an excellent, high quality manuscript.

Concern: The authors claim that macaques infected with an RhCMV genetically ablated for all 3 vFcR's are more readily able to clear the "plasma viremia," and their data is certainly consistent with this interpretation. However, the authors are measuring viral DNA in the plasma, DNA-emia, not necessary viremia / infectious or even non-infectious virions, per se. CMVs are widely reported to be "highly cell associated in vivo," with cell-free virions being scarce outside of saliva and urine, etc. Albeit that those observations are being made with viruses carrying intact vFcR loci, is it possible that the "plasma viremia" being detected is mostly non-infectious viral DNA and that the measurement of more rapid clearance of the "plasma DNA-emia" is indicating something other than more efficient antibody-dependent adsorption of virions in triple vFcR deletion virus infected animals?

Minor comments.

(1) Line 124 and surrounding text + Fig 1 Legend. The authors need to make clear to readers that CD16 is FcγRIII. Although many readers will already know this, many will not. Adding a few words will rectify this to make the rationale for the experiment clearer to a wider readership. i.e., "Rh152/151 was able to directly interfere with the binding of rhesus FcγRIII (CD16).."

(2) Fig 1(a) and its write up in the Results section are a little confusing. The narration could be a little punchier and clearer. The overall point being shown is that Rh152/151 functionally echoes HCMV UL119/118 binding IgG1 to inhibit FcγRIII. I'm not seeing much difference at all between the results for permeabilized and non-permeabilized cells and maybe that ought to be succinctly explained. Also, should the labeling refer to "permeabilized" and "non-permeabilized (surface only)" conditions, rather than the current labels .. In other words, is the term "Intracellular" misleading if the condition is showing signal for both surface + internal binding/ capturing of IgG, right? Or is an interpretation being left unspoken / unwritten that intracellular pools of viral FcR protein non-functional for binding IgG? Otherwise, why doesn't permeabilization lead to an

increase in signal for full length Rh152/151? Wouldn't that be expected since the cytoplasmic tail of the full length protein contains endocytic recycling motifs? Ironically, the way the results for this are written, the reader can get distracted from the main point.

(3) There's a mismatch in Fig 2A.. the middle pair of panels (currently included as part of Fig 2A) showing intracellular Fcγ and intracellular anti-HA staining signals starts at HA-Rh06 (RL11B) leave out the first 6 samples analyzed in the upper panel for MFI, i.e., HA-CD99 to HA-Rh152/151 (68-1). This is a source of potential confusion because on Lines 165-166: there is no "expression level" result shown in the middle panels for the positive control Rh05, only Rh173. However, the language on lines 165-166 makes it sound like that result can be found in the lower panel of 2A, when it is not.

(4) Typo on line 468 (Discussion), a "not" is written where a "no" was intended "...supported by the observation that not difference in peak"

(5) Maybe line 477 should start with "Perhaps unexpectedly," instead of "Unexpectedly"... or something else? This is definitely 100% up to the authors (prestidigitation /slight of hand and straw men do have a place in writing), but it certainly seems that they'd absolutely expect T cell evasion functions to be doing the heavy lifting when it comes to enabling superinfection, not antibody evasion. Much of the work revealing some of the elegant T cell mechanisms / biology at play was published by some of the co-authors.

(6) Line 533, BamHI and PstI should be written as such (as per latter) not as BamH1 and Pst1.

(7) Line 598 : Methods section on Multistep growth curves appears to be missing some details. Were serial 10-fold (or other fold) dilution series made? This methodology on titration to determine FFU should describe that aspect of how units of infectivity were enumerated / determined.

Reviewer #2

(Remarks to the Author)

This manuscript by Otero and colleagues expands our knowledge of CMV vFcγRs using the rhesus model. This group has previously characterized a RhCMV vFcγR, Rh05, and has now defined 2 others, Rh152/151 and Rh173, all of which are orthologs of HCMV vFcγRs. The experiments appear rigorous and the data are well-presented. The subject is of great importance to the field and takes advantage of an increasingly informative animal model.

Like Rh05, they show that the other vFcγRs bind IgG and inhibit FcγR activation. They also present data indicating that these proteins are likely expressed on the membrane of infected cells and on the virion surface. Deletion of these vFcγRs does not appear to affect viral fitness in vitro but, when deleted, result in a shorter duration of viremia in a small number of animals during primary infection.

The mechanism for faster clearance of viremia is suggested to be due to an absence of IgG binding and inhibition of Fc-mediated activities by deletion mutant viruses. However, the evidence for this is indirect at best, as shown by the fact that this effect is negated among CD4-depleted animals. In my opinion, the conclusion that the difference in viremia duration is due to Fc activity--however plausible--is over-interpreted. As suggested, a stronger experiment would have been to deplete B cells, as the others themselves point out in the discussion. The inverse experiment would be passively transfer antibody from immune animals prior to primary infection, which one might expect to enhance the difference in viremia between mutant and wild type virus.

Interestingly, deletion mutant virus appeared capable of causing reinfection of seropositive animals, suggesting that evasion of Fc-mediated immunity is not required. This experiment was performed by injection of 5x10E6 PFU of virus IV in 2 animals, and the read-out was the development of T cell responses to the SIV pol insert. One wonders, however, if this relatively crude binary test might miss even a marked level of protection, if reinfection with vFcγR deletion mutants could be compared with wild type to determine the ID50 and/or the amount of resulting viral replication. This possibility is not trivial, since it has great importance for the development of vaccines and antibody therapies with respect to the role of Fc functions in preventing infection and reinfection. Thus, conclusions regarding the role of vFcγRs and Fc-mediated antibody activities in reinfection based on this experiment should be much more nuanced. Furthermore, the discussion around the relevance of Fc-mediated functions in natural and vaccine-mediated protection and in congenital transmission could be expanded somewhat, to explore the implications of this study's findings.

It is well noted that the importance of these proteins can only really be determined in vivo, and that these animal experiments are enormously difficult and expensive, which limits the number of conditions that can be tested. The authors should thus be congratulated on a valuable contribution to this area of study.

Reviewer #3

(Remarks to the Author)

Otero et al. have identified and characterized two new viral Fc receptors (vFcγRs) encoded by rhesus cytomegalovirus (CMV). Other herpesviruses, such as herpes simplex virus (HSV), varicella-zoster virus (VZV), and human CMV (HCMV), encode different vFcγRs, and their immune evasion functions have been primarily described in vitro experiments.

Otero et al. conducted experiments in which they infected rhesus macaques with RhCMV that had three vFcγRs deleted.

They found that the duration of plasma viremia was significantly reduced in immunocompetent rhesus macaques but not in CD4+ T cell-depleted animals. This discovery helps advance our understanding of how vFcRs evade humoral immune responses in the host.

The manuscript is well-written and utilizes numerous recombinant RhCMV viruses in the experiments, with HCMV used as a control. However, several questions could be further addressed to enhance the quality of the manuscript.

1. It remains elusive whether RhCMV only encodes three FcgRs. In fibroblast cells infected with RhCMV deleted with three vFcYRs, can rhesus macaques IgG still bind to the cell surface? This experiment would inform us the RhCMV might encode more than three vFc Rs.

2. The binding affinity of the three RhCMV vFcYRs, especially RhR05 and Rh173, for IgG or IgG isotypes is interesting. This information may help explain the functional differences of the RhCMV vFcYRs.

3. Figure 1A, the line pattern representing the cell surface and intracellular staining of Rh152/151 vFcYR looks similar or the same. The authors may need to verify or pay attention to this result.

4. Lines 162-163, "However, as these genes lack either a signal peptide or a transmembrane domain in their original sequence, they are unlikely to encode viable vFcYR candidates." This statement may be inappropriate; the genes lacking a transmembrane domain in their original sequence will likely encode soluble vFcYR candidates.

5. The RhCMV vFcYRs were co-localized with the endosomal marker (EEA1) or the lysosomal marker (LAMP-1). It is currently unknown whether the RhCMV vFcYRs can bind IgG under acidic pH conditions.

6. The RhCMV may infect myeloid cells such as macrophages or dendritic cells. The infection of antigen-presenting cells may confer vFcYRs a different role in processing immune complexed antigens to influence CD4 and CD8 T cells. The authors may need to discuss this aspect.

Minors:

1. Line 76, what does the "HIG" mean?

2. Figure 2A, is the intracellular staining without permeabilization? ...for intracellular IgG-binding (left panel)? The left panel is labeled as "Cell Surface".

3. For non-CMV readers, the authors might need to define the pentameric complex (PC).

4. Figure 3E is crucial, but it is complex to comprehend. The authors should consider revising the Figure legend or illustrations to include more explanatory details.

5. There was no data to confirm CD4+ T-cell depletion in the CD4 mAb-treated rhesus macaques.

6. The sentences in the Lines 455-456, line 468 need to be rephrased.

7. Hela should be HeLa in numerous locations.

8. A rationale for using the T2A peptide (sequence?) could be given in a short sentence, although a reference was cited.

9. The CEO is in the funding section, but the CO is in the author's contribution. Keep it consistent.

10. Line 609, PBA? Line 786, 3 SDs?

Version 1:

Reviewer comments:

Reviewer #1

(Remarks to the Author)

The authors have addressed all my concerns. I congratulate them on a strong manuscript addressing an important and genuinely interesting area of research.

Jeremy P. Kamil, Ph.D.
University of Pittsburgh School of Medicine, USA

Reviewer #2

(Remarks to the Author)

This reviewer's concerns have been adequately addressed.

Reviewer #3

(Remarks to the Author)

I commend the authors for addressing the comments raised; I have no further comments.

Reviewer #4

(Remarks to the Author)

In this excellent study, Otero, Petkova, Ebermann et al investigate the impact of the Fc-gamma receptors encoded by cytomegalovirus on infection and pathogenesis. They identified and characterized Rh05, Rh152/151 and Rh173 as the complete set of vFcyRs encoded by rhesus CMV (RhCMV) and they thoroughly characterized these Fc-gamma receptors. They report that each of these proteins displays functional similarities to their prospective HCMV orthologs with respect to antagonizing host FcyR activation in vitro. Further, they infected RhCMV-naïve rhesus macaques with vFcyR-deleted RhCMV and they report that these infections presented peak plasma DNAemia levels and anti-RhCMV antibody responses comparable to wildtype infections. They also report that Rh infection with vFcyR-deleted RhCMV are characterized by a significantly shortened duration of plasma DNAemia was significantly shortened in immunocompetent, but not in CD4+ T cell-depleted Rh. Overall, they concluded that vFcyRs can prolong lytic replication during primary infection by evading virus-specific adaptive immune responses, particularly antibodies.

The paper is very well crafted, described studies are of importance and of potentially high impact. The animal studies are well designed and meaningful.

This is a resubmission and the authors were very responsive to the previous critiques.

I only have several comments/suggestions related to the CD4+ T cell depletion studies:

- a figure showing the efficacy and, most importantly, the duration of the CD4+ T cell depletion after the administration of a single dose of depleting antibody is necessary.
- also, in the figure showing the dynamics of Rh-CMV infection in CD4+ T cell-depleted animals, the authors should add an average curve of the dynamics of viral loads in undepleted animals, for meaningful comparisons.

Version 2:

Reviewer comments:

Reviewer #4

(Remarks to the Author)

The authors addressed all my concerns. The manuscript is ready for publication

Reviewer #1 (Remarks to the Author)

This study by Claire Otero and co-authors addresses the functional roles of cytomegalovirus encoded Fc binding proteins, aka "viral FcR's" during rhesus CMV (RhCMV) infection of macaques. Interestingly, HCMV, and as this study shows, RhCMV each encode multiple distinct FcR's. The literature suggests that each of the HCMV vFcRs possesses unique/ non-overlapping activities which arguably act with synergism to block antibody dependent effector functions such ADCC and ADP. Here, Otero et al. characterize the in vivo phenotypic outcome / impact on viral fitness during natural infection when all three RhCMV vFcR are genetically ablated, finding that virus lacking all 3 RhCMV vFcR exhibits a shortened plasma viremia but that the deletion virus is otherwise able to infect previously seroconverted animals. Thus the vFcR-triple null RhCMV still is able to defy "vector immunity," a feature of CMVs that relies T cell evasion functions. The authors also provide ex vivo characterization addressing how these viral molecules dampen cellular FcR activity, and evaluating their sub cellular localization and trafficking within infected cells. The authors had previously reported that Rh05 is a vFcR, and here they start out by identifying RhCMV counterparts for two additional HCMV vFcR's that to-date had not yet been formally identified in RhCMV, at least not beyond recognition of a positional homolog ORF. They find via screening of RL11 family members that Rh173 is a vFcR. The authors also observe that Rh13.1, which sequence data suggest is the RhCMV paralog of the HCMV vFcR RL13 does not in fact exhibit IgG binding activity in their hands.

Overall this is an excellent, high quality manuscript.

Concern: The authors claim that macaques infected with an RhCMV genetically ablated for all 3 vFcR's are more readily able to clear the "plasma viremia," and their data is certainly consistent with this interpretation. However, the authors are measuring viral DNA in the plasma, DNA-emia, not necessary viremia / infectious or even non-infectious virions, per se. CMVs are widely reported to be "highly cell associated in vivo," with cell-free virions being scarce outside of saliva and urine, etc. Albeit that those observations are being made with viruses carrying intact vFcR loci, is it possible that the "plasma viremia" being detected is mostly non-infectious viral DNA and that the measurement of more rapid clearance of the "plasma DNA-emia" is indicating something other than more efficient antibody-dependent adsorption of virions in triple vFcR deletion virus infected animals?

We thank the reviewer for pointing out that viral DNA in the plasma could have an alternative interpretation and we have altered our manuscript and our figures to present our results with more caution in response to the reviewer's suggestion. This includes using the term DNAemia and a change in wording [e.g. line 631, 468-469].

Minor comments.

(1) Line 124 and surrounding text + Fig 1 Legend. The authors need to make clear to readers that CD16 is FcγRIII. Although many readers will already know this, many will not. Adding a few words will rectify this to make the rationale for the experiment clearer to a wider readership. i.e., "Rh152/151 was able to directly interfere with the binding of rhesus FcγRIII (CD16)."

We have added this information to the manuscript when we first discuss the human FcγRs and we more specifically mention that we are working with CD16A F176V [line 134]. We have

furthermore updated the “CD16 flow cytometry binding assay” section in the materials and methods section as well to be consistent with the rest of the manuscript.

(2) Fig 1(a) and its write up in the Results section are a little confusing. The narration could be a little punchier and clearer. The overall point being shown is that Rh152/151 functionally echoes HCMV UL119/118 binding IgG1 to inhibit FcγRIII. I’m not seeing much difference at all between the results for permeabilized and non-permeabilized cells and maybe that ought to be succinctly explained. Also, should the labeling refer to “permeabilized” and “non-permeabilized (surface only)” conditions, rather than the current labels. In other words, is the term “Intracellular” misleading if the condition is showing signal for both surface + internal binding/ capturing of IgG, right? Or is an interpretation being left unspoken / unwritten that intracellular pools of viral FcR protein non-functional for binding IgG? Otherwise, why doesn't permeabilization lead to an increase in signal for full length Rh152/151? Wouldn't that be expected since the cytoplasmic tail of the full length protein contains endocytic recycling motifs? Ironically, the way the results for this are written, the reader can get distracted from the main point.

We agree with the reviewer that the wording we have used to describe our results might be confusing. The interpretation that we are comparing permeabilized and non-permeabilized cells is correct and we have edited the manuscript and the above mentioned figure to better convey our expected and observed results, which is that we do not see a difference between permeabilized and non-permeabilized transfected with the individual vFcγRs, as the removal of the sorting signal in the cytoplasmic tail will increase membrane stability and consequentially reduce subsequent degradation [line 124-125 + 129 + Fig 1 legend + Fig 1A].

(3) There's a mismatch in Fig 2A.. the middle pair of panels (currently included as part of Fig 2A) showing intracellular Fcγ and intracellular anti-HA staining signals starts at HA-Rh06 (RL11B) leave out the first 6 samples analyzed in the upper panel for MFI, i.e., HA-CD99 to HA-Rh152/151 (68-1). This is a source of potential confusion because on Lines 165-166: there is no "expression level" result shown in the middle panels for the positive control Rh05, only Rh173. However, the language on lines 165-166 makes it sound like that result can be found in the lower panel of 2A, when it is not.

We apologize for the confusion, but the two panels do not show the same set of samples. The upper panel depicts primary data comparing the previously characterized HCMV and here identified RhCMV vFcγRs to all RhCMV encoded RL11 family members including Rh173. In the lower panel we subsequently focused on just the RL11 family members using HCMV UL119/118 as a positive control to exclude general issues with the expression levels of the cloned RhCMV proteins. We did notice that we originally described the HCMV UL119/118 as containing a CD4 tail. That description is incorrect, and we have altered our manuscript to reflect the proper description of the UL119/118 construct (non-CD4-tailed). We have additionally changed the labeling of our presented figures according to the reviewer's suggestion to now read “permeabilized” and “non-permeabilized”. [line 178-181 + Fig 2A legend + Fig2A]

(4) Typo on line 468 (Discussion), a “not” is written where a “no” was intended “...supported by the observation that not difference in peak”

We thank the reviewer for pointing this out. The error was corrected [line 606]

(5) Maybe line 477 should start with “Perhaps unexpectedly,” instead of “Unexpectedly”... or something else? This is definitely 100% up to the authors (prestidigitation /slight of hand and straw men do have a place in writing), but it certainly seems that they’d absolutely expect T cell evasion functions to be doing the heavy lifting when it comes to enabling superinfection, not antibody evasion. Much of the work revealing some of the elegant T cell mechanisms / biology at play was published by some of the co-authors.

We appreciate the comment made by this reviewer as it indicates a deeper understanding of this field in general and our work in particular. Our original wording was slightly different, but we had to alter it to comply with the word limit set by the journal. We have reversed this decision as we feel that it better describes the data and our interpretation of it as suggested by the reviewer. [line 618]

(6) Line 533, BamHI and PstI should be written as such (as per latter) not as BamH1 and Pst1.

We changed the names according to the reviewer’s suggestion. [line 693]

(7) Line 598 : Methods section on Multistep growth curves appears to be missing some details. Were serial 10-fold (or other fold) dilution series made? This methodology on titration to determine FFU should describe that aspect of how units of infectivity were enumerated / determined.

The focus forming assay (FFA) used to determine the amount of infectious virus in the samples harvested during the multistep growth curved uses 1:5 serial dilutions. We have added this and other relevant information to the “Multistep growth curve” paragraph in the materials and methods section.

Reviewer #2 (Remarks to the Author)

This manuscript by Otero and colleagues expands our knowledge of CMV vFcyRs using the rhesus model. This group has previously characterized a RhCMV vFcyR, Rh05, and has now defined 2 others, Rh152/151 and Rh173, all of which are orthologs of HMCV vFcyRs. The experiments appear rigorous and the data are well-presented. The subject is of great importance to the field and takes advantage of an increasingly informative animal model. Like Rh05, they show that the other vFcyRs bind IgG and inhibit FcyR activation. They also present data indicating that these proteins are likely expressed on the membrane of infected cells and on the virion surface. Deletion of these vFcyRs does not appear to affect viral fitness in vitro but, when deleted, result in a shorter duration of viremia in a small number of animals during primary infection. The mechanism for faster clearance of viremia is suggested to be due to an absence of IgG binding and inhibition of Fc-mediated activities by deletion mutant viruses. However, the evidence for this is indirect at best, as shown by the fact that this effect is negated among CD4-depleted animals. In my opinion, the conclusion that the difference in viremia duration is due to Fc activity--however plausible--is over-interpreted. As suggested, a stronger experiment would have been to deplete B cells, as the others themselves point out in the discussion. The inverse experiment would be passively transfer antibody from immune animals

prior to primary infection, which one might expect to enhance the difference in viremia between mutant and wild type virus.

We are grateful for the reviewer's comments, and we appreciate the reviewer's insight and understanding of our here presented project. We agree that additional in vivo experiments would have strengthened our manuscript, but these additional experiments in RhCMV seronegative NHPs have to be planned very judiciously, as they are a limited resource. Hence, we will commence further experiments assessing the role of B cell depletion and hyperimmunoglobulin transfer (from seropositive RM to seronegative RM) on the viral replication kinetics in the near future when we have secured additional funding. For now, we believe that our interpretation of the data is sufficiently cautious as we refrain from using definitive language. We would also like to draw the reviewer's attention to the data presented in Fig. 5C/D, where we confirmed that there was no marked difference in the quality of the IgG response towards the FL-RhCMV WT or the vFcyR deleted recombinant. Combined with the results generated in CD4-depleted dams and our in vitro assessments in Fig. 1/2/4, we believe that our interpretation is in line with our data and confirms our hypothesis that deletion of the vFcyRs does not have a significant impact the humoral immune response itself, but that the lack of vFcyR expression rescues an efficient IgG-Fc mediated effector response.

Interestingly, deletion mutant virus appeared capable of causing reinfection of seropositive animals, suggesting that evasion of Fc-mediated immunity is not required. This experiment was performed by injection of 5×10^6 PFU of virus IV in 2 animals, and the read-out was the development of T cell responses to the SIV pol insert. One wonders, however, if this relatively crude binary test might miss even a marked level of protection, if reinfection with vFcyR deletion mutants could be compared with wild type to determine the ID50 and/or the amount of resulting viral replication. This possibility is not trivial, since it has great importance for the development of vaccines and antibody therapies with respect to the role of Fc functions in preventing infection and reinfection. Thus, conclusions regarding the role of vFcyRs and Fc-mediated antibody activities in reinfection based on this experiment should be much more nuanced.

The points raised by the reviewer are interesting but experimentally challenging. One important factor to consider is, that we are using outbred rhesus macaques to perform these studies, so the animal-to-animal variability is enormous. Hence, to detect rather minute changes that are not black and white, would require substantially larger cohort sizes that would make these experiments exponentially more expensive and complicated. While we fully agree that the answer to the question posed by the reviewer is important and worth exploring, we do currently not have the budget to perform additional in vivo studies, yet we are planning to revisit this point during future vaccine development projects using our model. For our here presented manuscript, we hope that our binary readout will suffice to support our important conclusion that deletion of the vFcyRs does not preclude superinfection of CMV seropositive animals.

Furthermore, the discussion around the relevance of Fc-mediated functions in natural and vaccine-mediated protection and in congenital transmission could be expanded somewhat, to explore the implications of this study's findings.

We have included a more detailed discussion of this point in the revised version of the manuscript [line 659-662]

It is well noted that the importance of these proteins can only really be determined in vivo, and that these animal experiments are enormously difficult and expensive, which limits the number of conditions that can be tested. The authors should thus be congratulated on a valuable contribution to this area of study.

We are grateful for the reviewer's supportive remarks and appreciation of our work.

Reviewer #3 (Remarks to the Author)

Otero et al. have identified and characterized two new viral Fc receptors (vFcγRs) encoded by rhesus cytomegalovirus (CMV). Other herpesviruses, such as herpes simplex virus (HSV), varicella-zoster virus (VZV), and human CMV (HCMV), encode different vFcγRs, and their immune evasion functions have been primarily described in vitro experiments. Otero et al. conducted experiments in which they infected rhesus macaques with RhCMV that had three vFcγRs deleted. They found that the duration of plasma viremia was significantly reduced in immunocompetent rhesus macaques but not in CD4+ T cell-depleted animals. This discovery helps advance our understanding of how vFcRs evade humoral immune responses in the host. The manuscript is well-written and utilizes numerous recombinant RhCMV viruses in the experiments, with HCMV used as a control. However, several questions could be further addressed to enhance the quality of the manuscript.

1. It remains elusive whether RhCMV only encodes three FcγRs. In fibroblast cells infected with RhCMV deleted with three vFcγRs, can rhesus macaques IgG still bind to the cell surface? This experiment would inform us the RhCMV might encode more than three vFcγRs.

We appreciate the reviewer's comment as it reflects our own initial concerns, but we would like to refer the reviewer to the Fc pull-down experiment presented in Fig.5D. In this experiment we can demonstrate that no further IgG binding glycoproteins can be detected in cells infected with the triple-deleted FL-RhCMV. Granted, it is of course possible that we have missed additional proteins due to their intracellular location and their lack of glycosylation (which should not be relevant for the RL11 family as demonstrated in Fig. 2). As our identified RhCMV vFcγR are orthologs of characterized HCMV proteins, we expect to have identified at least the most dominant, if not all proteins with Fcγ binding activity, but we cannot exclude that other family members might be identified in the future. We have added a more precise conclusion to our data to the results section [line 395-397] and to the discussion [line 584-587]. To address this point as suggested by the reviewer, we performed flow cytometry analysis on FL-RhCMV (or vFcγR deletion mutants thereof) infected TRF cells. Probing for IgG binding using human IgG-Fc (PE-TexasRed dye, Rockland) or rhesus CD4-specific rhesusized IgG1 (as in the manuscript and detected via total-IgG-specific PE labeled secondary antibody) revealed that in the absence of vFcγRs, no notable IgG binding over the mock control was detected.

- Human IgG-Fc probing: mock 242 MFI vs. vFcγR deleted FL-RhCMV 294 MFI.

- Rhesus IgG1 probing: mock 117 MFI vs. vFcR deleted FL-RhCMV 131 MFI.

TRF infected at MOI=1, 48h.
gated on total TRF. 10.000 counts

mock ■
VID-1256 (FL-RhCMV) ■
VID-1271 (Δ Rh05 Δ Rh152/151 Δ Rh173) ■

2. The binding affinity of the three RhCMV vFcgRs, especially RhR05 and Rh173, for IgG or IgG isotypes is interesting. This information may help explain the functional differences of the RhCMV vFcgRs.

We agree that examining the binding affinities of the identified vFcyRs would be interesting, especially in comparison to their HCMV orthologs for which such an assessment was performed (Sprague et al. J Virol 2008). Similarly, examining differences in binding of RhCMV encoded vFcyR to individual IgG subclasses would be of scientific interest, alas, we feel that this information is of more importance for the study of HCMV vFcyRs as they will be the actual vaccine targets while the RhCMV orthologs are only the *in vivo* model for vaccine development. Thus, we believe that additional characterization of the RhCMV vFcyRs would not add to our current study, but could be reconsidered for future *in vivo* studies.

3. Figure 1A, the line pattern representing the cell surface and intracellular staining of Rh152/151 vFcgR looks similar or the same. The authors may need to verify or pay attention to this result.

This point raised by this reviewer has been addressed in response to Reviewer#1.

4. Lines 162-163, "However, as these genes lack either a signal peptide or a transmembrane domain in their original sequence, they are unlikely to encode viable vFcyR candidates." This statement may be inappropriate; the genes lacking a transmembrane domain in their original sequence will likely encode soluble vFcyR candidates.

We agree that this is theoretically possible and interesting, especially as there are cytosolic IgG binding host molecules such as TRIM21. However, the data we have generated in this study indicates that RhCMV encodes for no further vFcyRs as demonstrated by our pull down experiment from infected cells presented in Fig. 4D. We would expect a secreted molecule to be glycosylated and therefore show up in our samples after de-glycosylation. An intracellular, non-secreted IgG binding molecule would most likely not exert an effect on host FcRs. We have altered our manuscript and now discuss this point in more detail [line 178-185].

5. The RhCMV vFcgRs were co-localized with the endosomal marker (EEA1) or the lysosomal marker (LAMP-1). It is currently unknown whether the RhCMV vFcgRs can bind IgG under acidic pH conditions.

The reviewer is right, it remains unknown for the RhCMV vFcyRs whether a change in pH would affect the IgG binding affinity. However, there are prior studies characterizing the HCMV vFcyRs, demonstrating that IgG binding is pH independent (Sprague et al. J Virol 2008). We

do not expect pH-dependency to affect the mechanisms discussed in this study, as vFcγR activity on the cell surface is assessed. Of note, in the mentioned subcellular localization experiments, we do not expect the reviewer's concern to interfere with our experiments, as we specifically detect these molecules using tag-specific antibodies from another species and show that there is no non-specific binding in Fig. S4.

6. The RhCMV may infect myeloid cells such as macrophages or dendritic cells. The infection of antigen presenting cells may confer vFcγRs a different role in processing immune complexed antigens to influence CD4 and CD8 T cells. The authors may need to discuss this aspect.

This is an interesting point, which we have now included into our discussion [line 624-629]. To address this question properly will require a more focused study of the T cell response and additional animals which will require additional funding.

Minors:

1. Line 76, what does the "HIG" mean?

HIG refers to hyper immunoglobulin which we have now defined in line 82.

2. Figure 2A, is the intracellular staining without permeabilization? ...for intracellular IgG-binding (left panel)? The left panel is labeled as "Cell Surface".

This question was raised by reviewer#1 above and we have changed our wording to "permeabilized" and "non-permeabilized".

3. For non-CMV readers, the authors might need to define the pentameric complex (PC). We have added a reference and an explanation in line 248-249.

4. Figure 3E is crucial, but it is complex to comprehend. The authors should consider revising the Figure legend or illustrations to include more explanatory details.

For clarity, we changed the y-axis to "mRNA copy ratio (RhCMV/GAPDH)". We also added information on the kinetic classes into the graph.

5. There was no data to confirm CD4+ T-cell depletion in the CD4 mAb-treated rhesus macaques.

The procedure was published in Bialas et al. PNAS 2015 and CD4+-depletion was confirmed by Alice Tarantal (UC Davis). We added information on the procedure to the manuscript in line 477-478 and in the MM section with additional information on the procedures and RMs used. Alice Tarantal was added as author. The procedure for CD4-depletion was as follows: Blood samples were collected prior to CD4 antibody depletion, at the 0 timepoint (immediately prior to RhCMV administration), then weekly for 12 weeks post-inoculation. CD4 T cell counts were assessed by flow cytometry prior to any interventions, then CD4 depletion was verified by flow cytometry from the blood sample collected immediately prior to RhCMV inoculation. In all cases, the dams showed complete depletion of CD4+ T cells and the levels of these cells were

monitored in all blood samples collected from these animals (and all animals on the study) during the 12-week evaluation period.

6. The sentences in the Lines 455-456, line 468 need to be rephrased.

We have made alterations to our manuscript to make these sentences more understandable. line 580-582 and line 608.

7. Hela should be HeLa in numerous locations.

We have made the suggested edit across our manuscript.

8. A rationale for using the T2A peptide (sequence?) could be given in a short sentence, although a reference was cited.

We thank the reviewer for pointing out that this was only explained in the Methods section and we have thus added an explanation to the reference in the main text in line 130-131.

9. The CEO is in the funding section, but the CO is in the author's contribution. Keep it consistent.

We thank the reviewer for identifying this conflict and we have made changes to keep the use of initials consistent.

10. Line 609, PBA? Line 786, 3 SDs?

PBA refers to PBS with 2% BSA. SD is a commonly used abbreviation for standard deviation that we failed to define. We have added definitions for both abbreviations to our manuscript. line 772 and line 1079.

Reviewer #1 (Remarks to the Author):

The authors have addressed all my concerns. I congratulate them on a strong manuscript addressing an important and genuinely interesting area of research.

Jeremy P. Kamil, Ph.D.

University of Pittsburgh School of Medicine, USA

Thank you, Jeremy, for helping us improve our manuscript.

Reviewer #2 (Remarks to the Author):

This reviewer's concerns have been adequately addressed.

We would like to thank the reviewer for his/her thoughtful input.

Reviewer #3 (Remarks to the Author):

I commend the authors for addressing the comments raised; I have no further comments.

We are grateful for the reviewer's help and we are pleased that our modified manuscript fulfilled all the reviewer's requirement for publication.

Reviewer #4 (Remarks to the Author):

In this excellent study, Otero, Petkova, Ebermann et al investigate the impact of the Fc-gamma receptors encoded by cytomegalovirus on infection and pathogenesis. They identified and characterized Rh05, Rh152/151 and Rh173 as the complete set of vFcγRs encoded by rhesus CMV (RhCMV) and they thoroughly characterized these Fc-gamma receptors. They report that each of these proteins displays functional similarities to their prospective HCMV orthologs with respect to antagonizing host FcγR activation in vitro. Further, they infected RhCMV-naïve rhesus macaques with vFcγR-deleted RhCMV and they report that these infections presented peak plasma DNAemia levels and anti-RhCMV antibody responses comparable to wildtype infections. They also report that Rh infection with vFcγR-deleted RhCMV are characterized by a significantly shortened duration of plasma DNAemia was significantly shortened in immunocompetent, but not in CD4+ T cell-depleted Rh. Overall, they concluded that vFcγRs

can prolong lytic replication during primary infection by evading virus-specific adaptive immune responses, particularly antibodies.

The paper is very well crafted, described studies are of importance and of potentially high impact. The animal studies are well designed and meaningful.

This is a resubmission and the authors were very responsive to the previous critiques.

I only have several comments/suggestions related to the CD4+ T cell depletion studies:

-a figure showing the efficacy and, most importantly, the duration of the CD4+ T cell depletion after the administration of a single dose of depleting antibody is necessary.

We agree with the reviewer that this piece of information might be useful for the reader to judge the value of our here presented results, and we have hence added this information as Fig. S9 to the manuscript [line 403]. We also edited the Methods section with more detail

-also, in the figure showing the dynamics of Rh-CMV infection in CD4+ T cell-depleted animals, the authors should add an average curve of the dynamics of viral loads in undepleted animals, for meaningful comparisons.

We have included this reasonable request into our modified Fig. 5E.

Reviewer #4 (Remarks to the Author):

The authors addressed all my concerns. The manuscript is ready for publication

We are grateful for the reviewer's help and we are pleased that our modified manuscript fulfilled all the reviewer's requirement for publication.